# Transposable elements hitchhike on *Starships* across fungal genomes

Hanne Griem-Krey [1], Júlia de Fraga Sant'Ana [2], Ursula Oggenfuss[3], Yohana Porto Calegari-Alves[2], Ana Luiza Marques[2], Markus Berger[4], Lucélia Santi [2], Walter O. Beys-da-Silva [2] ✉ & Michael Habig [1] ✉

Horizontal transfer of transposable elements (TEs) is widespread in eukaryotes, driving genetic variation and often associated with bursts of TE activity. Here, we report a recent TE burst in the insect-pathogenic fungus *Metarhizium anisopliae*. The actively transposing TEs were likely introduced via hitchhiking on a so-called *Starship*, a class of large, horizontally transferable transposons. This TE burst likely triggered extensive structural reshuffling across all chromosomes, which was associated with loss of pathogenicity. Expanding our analysis to other fungi, we found that *Starship*-mediated horizontal transfer of TEs is a general phenomenon. Most (75%) of 522 reported *Starships* harbor TEs; many of which show evidence of a recent burst, in some cases likely starting from the TE copies on the *Starship* itself. A high fraction of TEs located on *Starships* also shows signatures of past horizontal transfer. Collectively, our results establish *Starships* as major vectors of horizontal TE transfer.

Transposable elements (TEs) play a critical role in shaping genomes, contributing significantly to their evolution and the adaptation of species. TEs are classified into two major classes based on their transposition mechanisms: Class I retrotransposons, which transpose via an RNA intermediate, and Class II DNA transposons, which move through a DNA intermediate[1]. As mobile genetic elements, TEs can dramatically alter genome architecture, influence gene expression, and create genomic plasticity that enables species to adapt to changing environments. TEs can drive structural genome variation not only via insertions into different sites in different individuals but also through processes such as homologous recombination between TE copies[2], leading to chromosomal rearrangements including translocations, inversions, duplications, and deletions[3–5]. For example, *Styx* (also known as REP9), a Class II transposon in the fungal wheat pathogen *Zymoseptoria tritici*, has been independently activated multiple times in distinct populations and is associated with structural variations, triggering several chromosomal rearrangements, including deletions, duplications, and chromosomal fusions[6–8]. Transposition events may also result in insertional mutations[9,10], alter gene expression, or create new protein functions through exon shuffling and TE domestication[11–15]. Additionally, TEs serve as sources of regulatory elements[16], spreading cis-regulatory sequences across the genome and reconfiguring gene regulatory networks[17]. TE expansions are often associated with structural variant (SV) accumulation, as seen in domesticated crops such as tomato, rice, wheat, and soybean[18–21]. Notably, TEs and SVs tend to cluster in certain genomic regions, although it remains unclear whether this colocalization is causal or simply reflects shared tolerance to mutagenic change. To control the potentially harmful effects of TEs, host genomes have evolved various defense mechanisms, including DNA methylation[22], histone modification and heterochromatin formation[23], RNA-silencing pathways[24], and in fungi, repeat-induced point mutation (RIP), which introduces high C-to-T mutation pressures in repetitive sequences to inactivate TEs[25]. The vast majority of TEs are considered transpositionally inactive, possibly due to these defense mechanisms, which have been theoretically proposed to ultimately eliminate TEs from most species - a paradox, given their persistent and widespread presence across many genomes[26].

Horizontal transfer of TEs (HTT) is increasingly recognized as a significant driver of genome evolution in eukaryotes and may enable

[1]Fungal Evolutionary Genetics, Christian-Albrechts University of Kiel, Kiel, Germany. [2]Faculdade de Farmácia; Centro de Biotecnologia, Federal University of Rio Grande do Sul (UFRGS), Porto Alegre/RS, Brazil. [3]Laboratory of Evolutionary Genetics, Institute of Biology, University of Neuchâtel, Neuchâtel, Switzerland. [4]Hospital de Clínicas de Porto Alegre, Porto Alegre/RS, Brazil. ✉e-mail: walter.beys@ufrgs.br; mhabig@bot.uni-kiel.de

TEs to escape a specific host genome's silencing mechanism[26]. While most documented cases have focused on plants[27,28] and animals[29–31], evidence for HTT in fungi remains comparatively limited[32–36]. A recent example includes the proposed horizontal transfer of TEs between rice- and wheat-infecting lineages of the fungal phytopathogen *Pyricularia oryzae*[37]. More broadly, a recent screen of 1,348 fungal genomes reported widespread HTT, with up to ~70% of the TE complement in some genomes potentially acquired horizontally; HTT events were particularly common in *Mucoromycotina*, *Sordariomycetes*, *Dothideomycetes*, and *Leotiomycetes*[38]. HTTs can involve functionally diverse transposons[39], including those capable of introducing new introns[40]. Although thousands of HTT events have been identified across eukaryotes[31,32,39], the mechanisms underlying these transfers remain poorly understood. Proposed vectors include viruses and extracellular vesicles[41–44]; for instance, LINE1 retrotransposons have been experimentally shown to be packaged in extracellular vesicles and transferred between cultured human cells[43], while other TEs have been detected within large DNA viruses infecting arthropods[41]. In fungi, introgression has been indicated in the horizontal transfer of TEs between *Saccharomyces cerevisiae* and *Saccharomyces paradoxus*[34,36]. Parasexuality, a process allowing genetic recombination independent of meiosis, has been observed in several fungal species and could theoretically introduce TEs into new genomic backgrounds[45,46]. Similarly, horizontal chromosome transfer (HCT), documented in various fungal pathogens, could transfer TE-rich chromosomes between strains[47–51]. However, experimental confirmation of TE movement via parasexuality or HCT is still sparse. In summary, although HTT appears to play an important role in fungal genome evolution, the underlying mechanisms remain poorly characterized and merit further investigation.

*Starships* are a recently characterized group of massive transposable elements exclusive to fungi within the *Pezizomycotina* subphylum[52–55]. Ranging from 15 to nearly 700 kilobase pairs in length[56], these elements are mobilized by a conserved tyrosine recombinase gene known as Captain, which contains a DUF3435 domain located at the 5' end of the element[57]. *Starships* frequently carry diverse cargo genes that encode traits conferring ecological and evolutionary advantages, such as virulence factors (e.g., the necrotrophic effector ToxA)[54,58], resistance to formaldehyde[59] and heavy metals[57], and biosynthetic gene clusters[58,60]. Typically found as single copies in fungal genomes, *Starships* have been implicated in promoting genome plasticity by contributing to large-scale structural variations, including chromosomal rearrangements and insertions into lineage-specific regions, as observed in species such as *Verticillium dahliae*[61], *Macrophomina phaseolina*[62], and *Aspergillus fumigatus*[63,64]. A defining feature of *Starships* is their capacity for horizontal gene transfer between fungal species, facilitating the rapid acquisition of adaptive traits. This phenomenon has been documented in both plant and human fungal pathogens, where near-identical *Starships* have been found across distinct taxa and are frequently associated with enhanced virulence or environmental resilience[57,61,65–67]. Recently, the horizontal transfer of two *Starships* was experimentally demonstrated through co-culturing experiments, in which strains containing *Starships* successfully transferred them to conspecific and heterospecific strains lacking the elements[68]. Notably, horizontal transfer occurred even between *Paecilomyces variotii* and *A. fumigatus*, two species that diverged approximately 100 million years ago[68,69]. These findings establish *Starships* as active vectors for eukaryotic horizontal gene transfer[68].

Although *Starships* are known to mediate genome rearrangements and horizontal gene transfer, their role in transferring other TEs and the impact of such events on recipient genomes remain largely unexplored since they have only been recently discovered. Here, we use the insect-pathogenic, asexual fungus *Metarhizium anisopliae*[70] as an informative model to investigate the origin and consequences of horizontally transferred TEs. The genus *Metarhizium* includes over 50 globally distributed species with varied plant associations, insect host ranges, and reproductive modes[70,71]. Some *Metarhizium* species are widely used as biocontrol agents against insect pests[70,72,73]. Even within the same species, *Metarhizium* isolates exhibit considerable phenotypic diversity[73–75]. To date, little is known about the extent or effectiveness of genome defenses against TEs in *Metarhizium*. Since RIP is considered to happen solely during the sexual stages[25], the mostly asexual life cycle of several *Metarhizium* species[70] will likely prevent RIP from acting against TEs in these species and may also explain the scarcity of reported RIP signatures in the genomes of *Metarhizium* spp.[76]. However, parasexuality and cell fusion during insect infection may occur at high frequency, may facilitate horizontal transfer of genetic information between different *Metarhizium* strains, and may indicate the importance of non-sexual mechanisms in generating genetic variation[48,77,78].

Here, we identified a specific case in which a large number of TEs were most likely introduced into one *M. anisopliae* strain via a *Starship*, subsequently expanded, and ultimately led to genome-wide rearrangements and loss of pathogenicity. We subsequently broadened our analysis to additional fungal taxa, revealing that most of the 522 analyzed *Starships* carry TEs, many of which include identical TE copies that have expanded within host genomes, even across species boundaries. These findings suggest that *Starships* are major vectors for the horizontal transfer and expansion of TEs in fungi.

## Results

### *M. anisopliae* strains NE and E6 differ in accessory chromosomes and TE composition

We characterized the genome organization of the *M. anisopliae* strains NE and E6 using PacBio HiFi sequencing (see Supplementary Data 1 for an overview). Sequence analysis yielded assemblies for both strains with chromosome-level resolution, identifying seven confirmed or eight putative core chromosomes, along with one accessory chromosome in E6 and two in NE (Fig. 1a; Supplementary Fig. 1). All assembled chromosomes in E6 possess telomeres at both ends, while in NE, three assembled chromosomes are capped at both ends, and an additional three have telomeres on only one end (Fig. 1a). Presence/absence polymorphism analysis across five previously published *M. anisopliae* isolates (Supplementary Fig. 1a) confirmed that chromosomes chrC, chrD, and chrE are accessory. These chromosomes also showed presence/absence variation across 19 published genomes of various *Metarhizium* species (Supplementary Fig. 1b), further supporting their accessory nature. The identity and size of the accessory chromosomes were validated by pulsed-field gel electrophoresis, followed by sequencing of the excised chromosomal bands (Supplementary Fig. 1c, d), which closely matched the sizes determined from the assemblies. In the NE strain, chrC and chrD measured approximately 1.6 Mb and 1.4 Mb, respectively, and a small core chromosome (chr8, ~2.3 Mb) was present that lacked a corresponding chromosomal band in strain E6, but later was shown to consist of rearranged sequences of E6 core chromosomes (see below). In contrast, E6 harbored chrE (~1.3 Mb). The genome of NE is 3.6 Mb larger than that of E6, primarily due to the additional accessory chromosome and an expansion of TEs, which increased from 3.3% in E6 to 9.8% in NE (Fig. 1d; Supplementary Data 1). While this expansion is partly explained by the higher TE content on the accessory chromosomes (Fig. 1b, d), elevated TE content was also observed on the core chromosomes (Supplementary Data 1). Consistent with previous findings[48], the three identified accessory chromosomes exhibited higher TE density and lower gene content. Additionally, genes located on accessory chromosomes displayed significantly different codon usage patterns (Fig. 1c), suggesting distinct evolutionary histories compared to those on the core chromosomes[79]. In summary, the genomes of the two *M. anisopliae* strains NE and E6 show comprehensive structural differences.

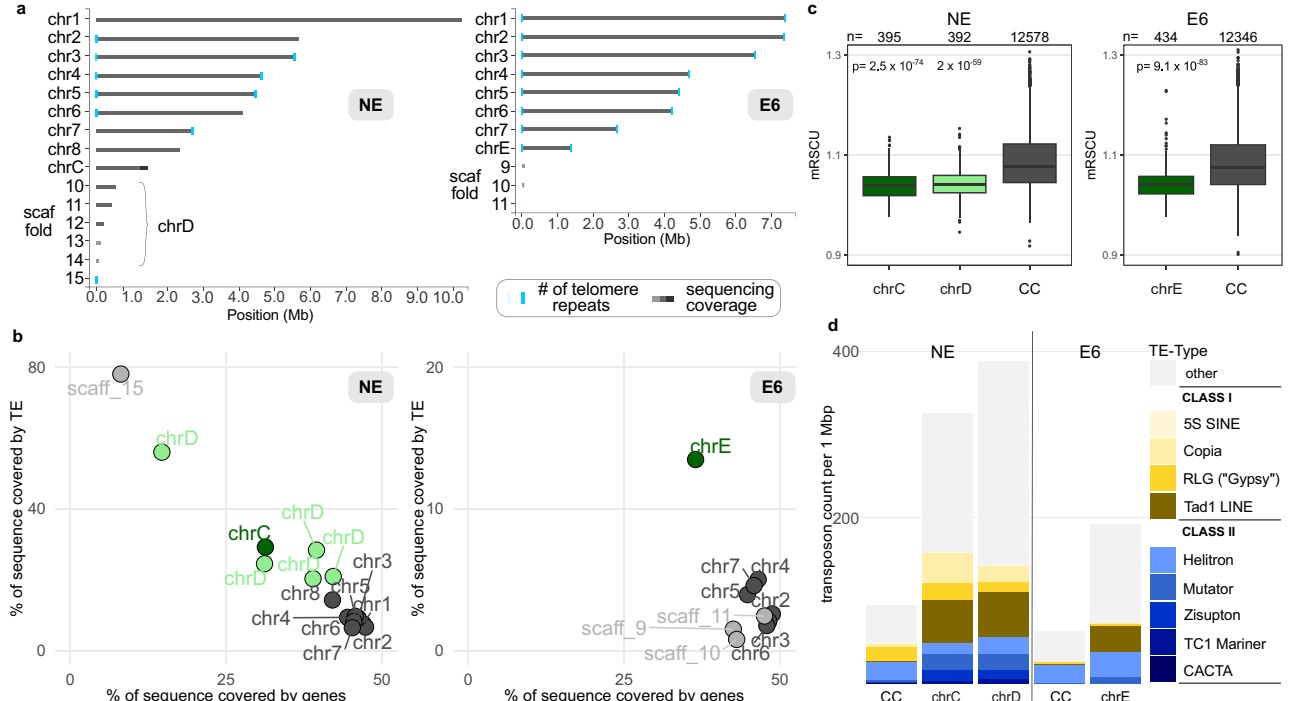

**Fig. 1 | *M. anisopliae* strains NE and E6 differ in genome organization, accessory chromosomes and TEs. a** Graphical representation of the genome assemblies for *M. anisopliae* strain NE (left) and E6 (right). In E6, all chromosomes have been fully assembled from telomere to telomere, and the NE assembly is also at near-chromosome level. **b** Distribution of genes and TEs between core and accessory chromosomes for strains NE (left) and E6 (right). The dot plots show the percentage of the sequence covered by TEs (Y-axis) and genes (X-axis) for core chromosomes (dark gray), accessory chromosomes (light and dark green), and unplaced scaffolds (light gray). Accessory chromosomes are distinct from core chromosomes by having higher TE content and lower gene content. **c** Genes on accessory chromosomes exhibit a different codon usage bias compared to genes on core

chromosomes. The mean gene-wise relative synonymous codon usage (RSCU) is shown for gene transcripts located on accessory and core chromosomes. n represents the number of individual transcripts of genes within the genomic compartment (chrC, chrD, chrE, or the core chromosomes). P-values from two-sided Wilcoxon rank-sum tests (compared to core chromosomes (CC)) are shown above each plot. Boxplots display the median (center line), the 25th and 75th percentiles (box bounds), and the minima and maxima via whiskers (1.5x IQR). Outliers are plotted as individual points beyond the whiskers to show the full data range. **d** TEs are overrepresented on accessory chromosomes, and their expansion is more pronounced in NE compared to E6, primarily involving class I (copy-and-paste) transposons.

## *M. anisopliae* strain NE has recently undergone extensive chromosome reshuffling

Synteny analysis of orthologous genes revealed a dramatic loss of macrosynteny between the NE and E6 strains (Fig. 2a). While E6 exhibits a high degree of macrosynteny conservation with chromosome-level assemblies of the more distantly related *Metarhizium robertsii* and *Metarhizium brunneum*, the comparison between the closely related NE and E6 genomes identified 37 major breakpoint regions (see Supplementary Data 2a, b). Of these, 28 are present in the fusion of sequences originally located on two different E6 chromosomes (termed "interchromosomal"), while the remaining nine joined non-syntenic regions from the same E6 chromosome, possibly representing large-scale inversions. For example, NE chromosome 1 is a mosaic of sequences syntenic to six different chromosomes in E6, and the newly formed core chromosome chr8 in NE is composed of two large segments syntenic to E6 chromosomes 5 and 7. Based on orthologous gene-based synteny, the breakpoint regions could be localized to intervals ranging from 412 to 54,879 bp in length. Phylogenetic analysis of 347 single-copy orthologous BUSCO gene sequences shared between NE, E6 and published *Metarhizium* genomes confirmed that the two strains are closely related and form a monophyletic clade within *M. anisopliae*, along with other published strains (Supplementary Fig. 2a, b). Furthermore, no substantial variation in SNP or small INDEL patterns (a total of 162,353 SNPs & small INDELs differentiate E6 from NE) was observed between NE and E6, and the distribution of these variants along the NE chromosomes did not correspond to the mosaic

synteny pattern (Supplementary Fig. 2c). Therefore, we exclude the possibility that the mosaic synteny in NE resulted from a hybridization event between more distantly related species. Instead, we conclude that the genome of NE has recently undergone major chromosomal rearrangements.

## Synteny-breaks are frequently associated with three distinct TEs
In 17 of the identified breakpoint regions, we detected individual TE sequences that belong to three different TE families: hAT-Restless, Helitron, or Mutator, in four, six, and seven regions, respectively (Fig. 2b). In contrast, no TE was detected in the remaining 20 breakpoint regions. Further nucleotide-level alignment of these breakpoint regions allowed us to pinpoint them more precisely. In many of the 17 TE-associated breakpoint regions, the TE was on both sides, or at least on one side, directly adjacent to a syntenic alignment (Fig. 2c; Supplementary Fig. 3a), suggesting that these TEs likely played a role in mediating genome rearrangement at these sites. For the breakpoint regions without an associated TE, we were able to localize the synteny disruption to within 10 base pairs in 17 of the 20 breakpoint regions. Although there was little sequence conservation among these breakpoint-adjacent regions, we observed a potential conserved motif shared among several of them (Supplementary Fig. 3b, c), but did not find any sequence similarity to known motifs (TEs or otherwise). Collectively, these results underscore that TEs may contribute to the formation of synteny breaks, while also indicating that additional, TE-independent mechanisms may underlie genome rearrangement at other sites.

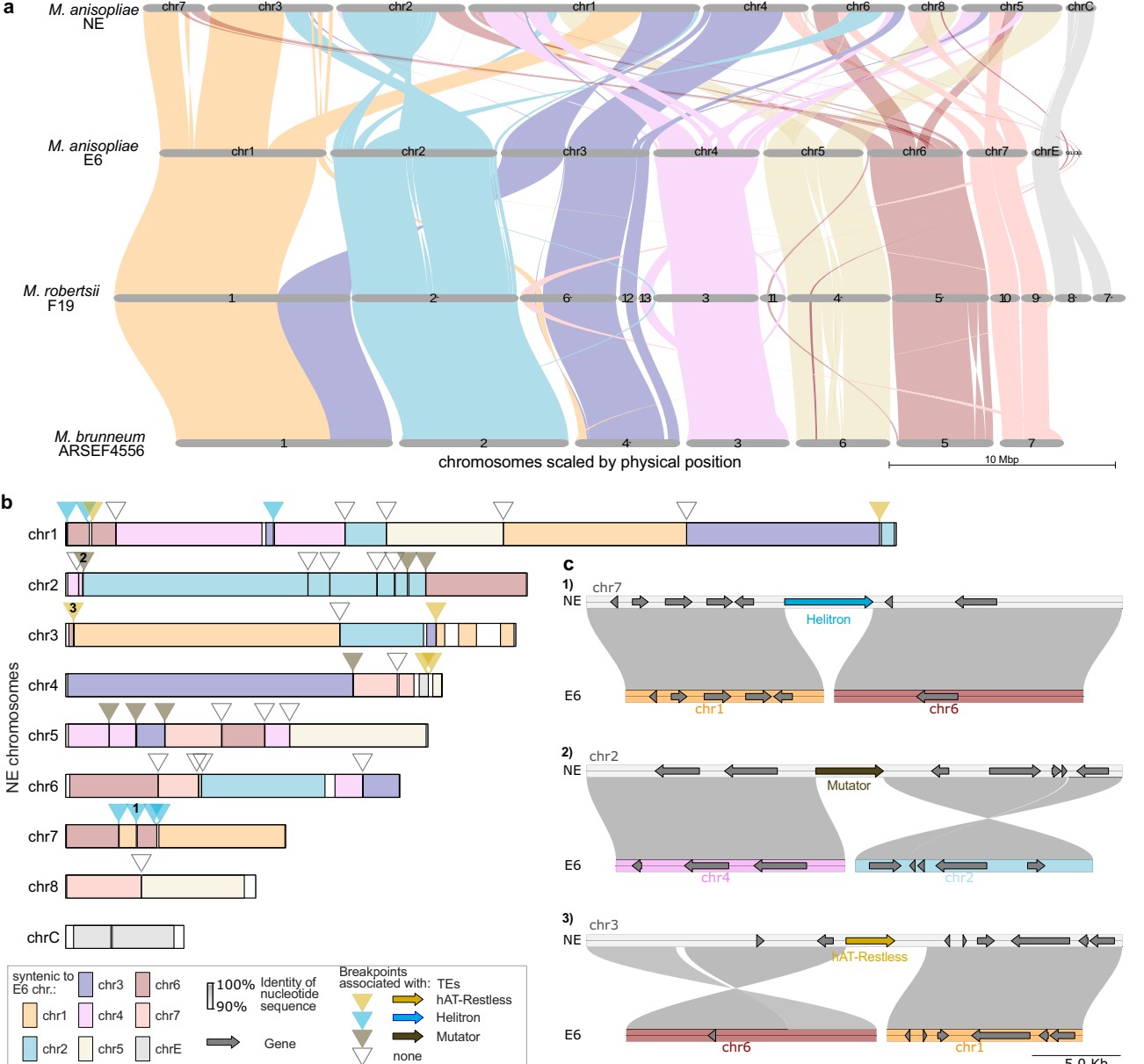

**Fig. 2 | *M. anisopliae* strain NE has recently undergone extensive structural genome rearrangements, with many breakpoints associated with three distinct TEs. a** Macrosynteny between *M. anisopliae* strain E6 and its related species, *M. robertsii* and *M. brunneum*, is highly conserved. In contrast, the *M. anisopliae* strain NE genome exhibits widespread rearrangements affecting all core chromosomes. **b** Depiction of chromosomes of NE colored according to the macrosyntenic relationships (based on gene synteny) with E6 chromosomes. Breakpoint regions are marked by triangles, colored to indicate the presence of one of three types of TEs: yellow for hAT-Restless, blue for Helitron, and brown for

Mutator. Open gray triangles denote breakpoints without these TEs. For clarity, chrD and scaffold_15, which do not contain breakpoints, are not shown. **c** Representative examples of nucleotide-level synteny at three breakpoint regions (1, 2, 3 as indicated in panel b), each associated with one of the three different TEs. These examples illustrate how TE-associated rearrangements connect sequences that are syntenic to regions on two different E6 chromosomes, which resulted in inter- and intra-chromosomal shuffling. For a complete overview of nucleotide synteny at all breakpoint regions see Supplementary Fig. 3. Please note that breakpoint regions associated with *Starships* are not included.

## The TE sequences were likely horizontally acquired and associated with a recent transposition burst

We detected a total of 196 near-identical copies of the three different TE families associated with structural rearrangements in the NE genome (Fig. 3a, b). The individual copies of each of these three TE families exhibited very little sequence variation between each other, with the majority being completely identical, indicating a very recent transposition burst (Fig. 3b–d). In most cases, this identity extended to the terminal inverted repeats (TIRs), which are 22 bp and 95 bp in length for the hAT-Restless and Mutator elements, respectively. These TIRs

are essential for the transposition of these elements, and their high sequence identity among numerous copies further supports a recent increase in the copy number of these functional transposons.

To investigate the origin of these recently expanded TE families, we analyzed their phylogenies, and in each case, the expanded TE family members formed a monophyletic group distinct from other members of the same TE superfamily in the NE genome. For example, the copies of the Helitron TE family that had increased in copy number (highlighted in the figure by the same background color) are clearly distinct from other Helitron TEs present in the NE genome (Fig. 3c).

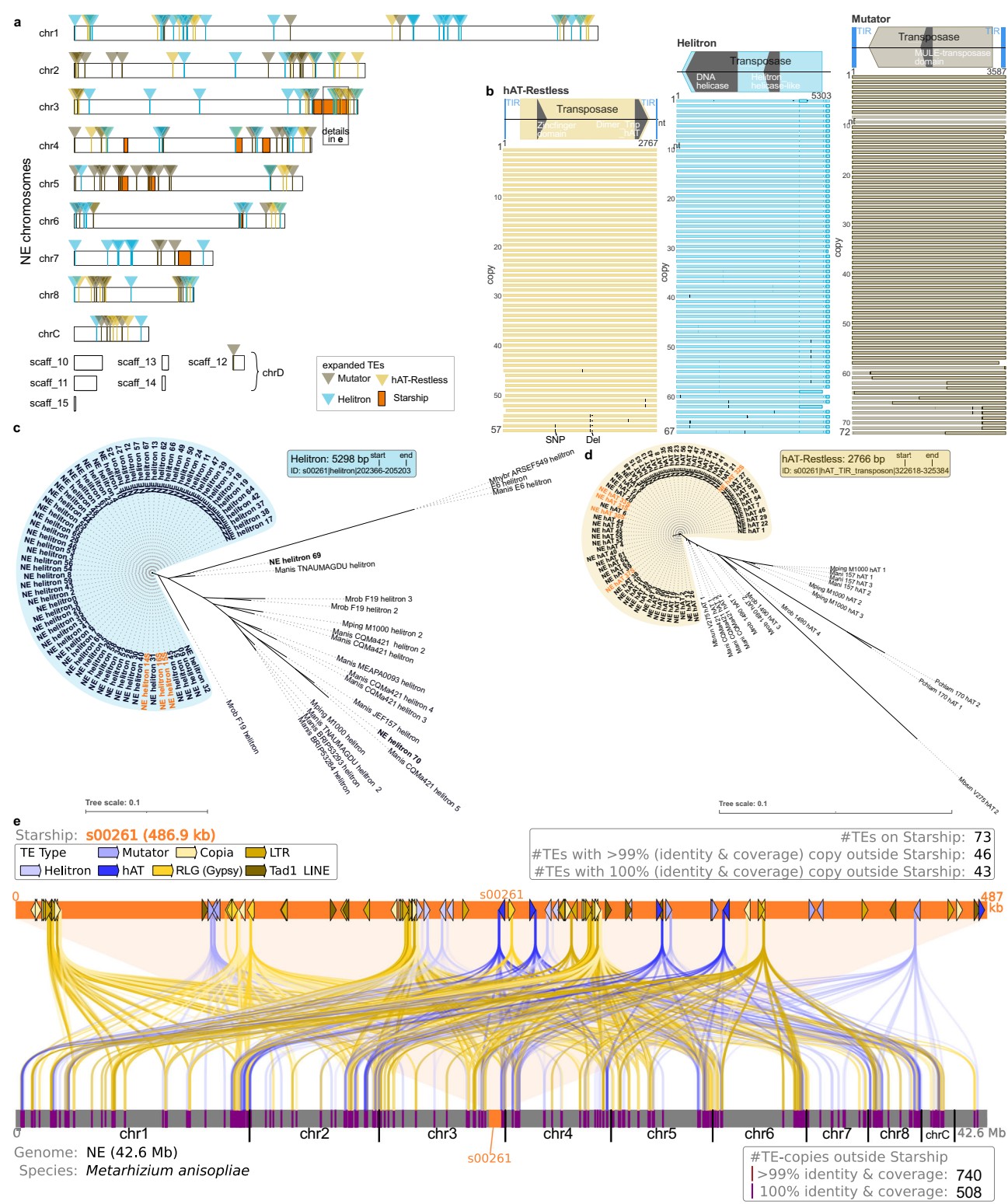

The phylogenetically closest Helitrons are found in a separate species, *M. robertsii*. Similarly, the copies of the hAT-Restless and the Mutator TEs that increased in copy number were monophyletic and distinct from other hAT-Restless and Mutator elements in the NE genome and most closely related to the respective TE elements in *M. brunneum* or *M. robertsii*, respectively (Fig. 3d; Supplementary Fig. 4a). We therefore conclude that TE families that increased in copy number did not originate within strain NE but were most likely acquired through horizontal transfer.

## A horizontally acquired *Starship* probably brought along TE sequences that subsequently underwent a burst of transpositions

The presence of three different types of DNA transposons that appear to have been concurrently horizontally acquired and contributed to genomic reorganization could be explained by the horizontal acquisition of larger genetic elements. Consequently, we searched for such elements and detected the presence of active *Starships* in the *M. anisopliae* strain NE, as well as in nine out of 19 isolates from different

**Fig. 3 | Three distinct TEs have recently expanded in *M. anisopliae* strain NE, and were likely horizontally acquired via *Starship* s00261, along with many additional TEs. a** Distribution of near-identical copies of Mutator, Helitron, and hAT-Restless TEs (depicted as yellow, blue, and brown triangles, respectively) across the chromosomes of the *M. anisopliae* strain NE. A total of 196 near-identical copies of these three TE families are present. Copies of all three TE families are located within *Starship* s00261 on chromosome 3 (*Starship* regions are orange). **b** Nucleotide alignment of copies of these TE families reveals only minor differences (SNPs & deletions) between the copies. For orientation, a schematic of their predicted functional domains is shown at the top. Phylogenetic trees of the Helitron (**c**) and hAT-Restless (**d**) TEs based on >80% identity and coverage copies among *Metarhizium* species. TEs from *M. anisopliae* strain NE are shown in bold,

those located within *Starship* s00261 in orange, with those copies depicted in part **b** of this figure having the same background color. Copies of the TE families that recently increased in copy number, including those from s00261, form monophyletic clades distinct from other members of the same TE family or superfamily present in *M. anisopliae* strain NE (see also Supplementary Fig. 4a–d for the phylogeny of additional TEs located on *Starship* s00261). Scale bars represent the expected substitutions per site. **e** Detailed map of *Starship* s00261 (orange, top) with the positions of annotated TEs (including fragments >100 bp) color-coded by TE superfamily. Bottom: location of identical copies (100% identity and coverage) of these s00261-TEs found elsewhere in the genome (purple). s00261 contains 73 TEs >100 bp, 43 have at least one fully identical copy outside s00261, totaling 508 identical copies across the genome.

*Metarhizium* species. Note that *Starship* detection presently requires at least two assemblies with different *Starship* positions, meaning that only relocated *Starships* are detectable[80]. We found that species within the *Metarhizium* genus harbor a high number of *Starships* (a total of 39) and associated tyrosine recombinase (YR) proteins (Supplementary Fig. 5a; Supplementary Data 3a, b). The *M. anisopliae* strain NE contains a total of nine *Starships*, spanning 1.6 Mb in total, and encoding 19 putative YR proteins (Supplementary Fig. 5a, b). The *Starships* in NE belong to four different families and are absent in *M. anisopliae* E6, which lacks transposed *Starships* but contains six putative YR genes (Supplementary Fig. 5c). Phylogenetic and synteny analyses of the Captains (putative YR genes, possibly involved in the excision and mobilization of the *Starship*) and the associated *Starships* in NE revealed that the Captains are phylogenetically diverse and distinct from the YR genes present in the E6 strain (Supplementary Fig. 5d). Interestingly, the Captains of four *Starships* in NE (s00260, s00265, s00266, s00268) exhibit high sequence similarity, which also extends to portions of their cargo (Supplementary Fig. 5e). However, the insertion sites of these four *Starships* show no synteny, indicating that the multiple similar *Starships* found in NE are the result of independent insertion events, rather than the amplification of a single ancestral element during genome reorganization. In conclusion, *Metarhizium* spp. can harbor multiple *Starships*, with up to nine elements found in a single genome (*M. anisopliae* strain NE). Notably, NE appears to have undergone a recent expansion of *Starships*, possibly through repeated horizontal acquisition events.

Moreover, we observed that a specific *Starship* (s00261), located on chromosome 3 in *M. anisopliae* strain NE, carries multiple copies of the TE families noted above - three Helitron, four hAT-Restless, and two Mutator elements - whose copy numbers have recently expanded. One possible explanation could be that these TE families may have entered NE via horizontal acquisition of this *Starship*. We therefore first tested whether *Starship* s00261 itself was horizontally acquired in NE. A *Starship*'s mobility depends on its Captain, the tyrosine recombinase, which in s00261 is encoded by gene g6294. This gene lacks a close phylogenetic counterpart in the E6 strain (Supplementary Fig. 5d) and we detected a presence/absence polymorphism for this gene across ten *Metarhizium* species (with a total of 20 isolates), with the closest phylogenetic orthologs in *Metarhizium humberi* (strain ESALQ1638) (Supplementary Fig. 6a, b). An Approximately Unbiased (AU) test[81] on the nucleotide alignment of the Captain and its homologs significantly rejected the species-tree topology inferred from 347 single-copy BUSCO orthologs (pAU = $1.5 \times 10^{-8}$). Together, the presence-absence pattern and the discordance between the gene tree and the species tree indicate that this Captain-encoding gene was likely acquired by horizontal transfer. Beyond the Captain, *Starship* s00261 encodes 123 additional cargo genes. Four of these are unique, with no homologs in the surveyed *Metarhizium* species. Among the remaining 119 genes with detectable homologs, 104 (86.7%) reject the BUSCO-based species-tree topology in AU tests, and many exhibit presence/absence polymorphism (Supplementary Fig. 6a). Collectively, these lines of evidence support the conclusion that *Starship* s00261 was

horizontally acquired into the NE strain, but the exact origin of this *Starship* is unknown.

We therefore next examined the TEs that are the cargo of the likely horizontally acquired *Starship* s00261. In total, 73 distinct TEs are present on *Starship* s00261 (Fig. 3e; Supplementary Data 4). These include the nine copies of the TE families noted above (Helitron, hAT-Restless, and Mutator). It is important to note that these TEs encompass full-length TEs as well as fragments greater than 100 bp, which may include solo-LTRs or products of nested insertions, for instance, all of which will subsequently be referred to as TEs. We first checked whether any other TEs on this *Starship* had recently expanded. We found that the majority (46 of the 73 TEs) had at least one copy elsewhere in the genome with >99% sequence identity and >99% coverage, which would indicate a recent expansion. Moreover, of these 46, 43 TEs had at least one identical copy (100% sequence identity across 100% of their length) outside the *Starship*, with 21 of these 43 being longer than 1000 bp, which would indicate a very recent expansion. These TEs include both DNA transposons and retrotransposons. In total, 740 copies (or 508 at 100% identity) of TEs located on *Starship* s00261 are dispersed throughout the NE genome (Fig. 3e). Further phylogenetic analysis of three highly expanded, larger (>1000 bp) TEs on *Starship* s00261 - two distinct RLG, formerly known as "Gypsy"[82], and one LTR element - revealed that the expanded copies cluster together phylogenetically and are distinct from other members of the same families or superfamily elsewhere in the genome. Both RLG ("Gypsy") retrotransposons have other members of the same TE family in the NE genome, but their closest phylogenetic relatives are found in *Metarhizium pinghaense* or *M. robertsii* (Supplementary Fig. 4b, d). In the case of the LTR retrotransposon, no close relative was found within the *Metarhizium* genus, we looked for closely related sequences beyond the *Metarhizium* genus and found the most closely related sequence in *Hirsutella rhossiliensis* (see Supplementary Fig. 4c), a soil-associated ascomycete that infects nematodes[83]. Consequently, all large TEs that have recently increased in copy number within the *M. anisopliae* strain NE and are located on *Starship* s00261 are phylogenetically distinct from members of the same TE superfamily present in NE or other *Metarhizium* spp. isolates. We therefore conclude that these large TEs were most likely acquired horizontally.

Was *Starship* s00261 the source or the sink of the expanded TEs? The presence of expanded TEs on the *Starship* could indicate that it acts as a source, a sink, or both for these expanded TEs. To disentangle this, we used the age of the TE as measured by sequence divergence as a proxy: older copies (i.e., those that have diverged more from their family consensus) are more likely to represent the source of an expansion. For each TE longer than 1000 bp, we collected all genome-wide copies with >99% identity and >99% coverage to that TE, built a consensus sequence for each resulting high-identity family, and measured the divergence of each copy to its family consensus. For each TE family, we then compared whether copies inside s00261 were, on average, older than copies outside s00261. We found no significant difference in divergence between copies inside *Starship* s00261 and those outside (Supplementary Fig. 7a). This likely reflects the very low

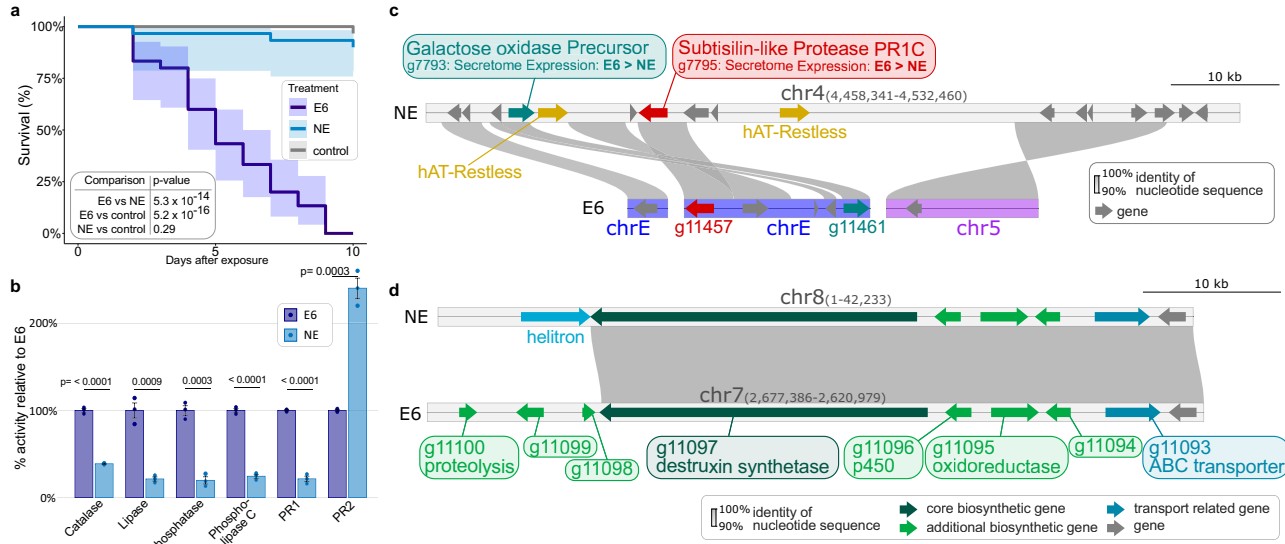

**Fig. 4 | Effect of structural variation on the pathogenicity and secretome of *M. anisopliae* strain NE. a** Kaplan-Meier survival curves of *R. microplus* ticks inoculated with *M. anisopliae* strain NE (light blue), strain E6 (dark blue), or a control (gray). Strain NE is less pathogenic than strain E6 and does not differ significantly from the control group (shaded areas: 95% confidence intervals of the mean). Data of three independent replicates with ten individuals/treatment each were pooled. Statistical significance was determined using log-rank test with Benjamini-Hochberg(BH)-adjustment for multiple testing. **b** Comparison of enzymatic activity between strains E6 and NE. Activities of each enzyme class were normalized to those of strain E6. Catalase, lipase, phosphatase, phospholipase C, and PR1 exhibited significantly reduced activity in strain NE, while PR2 activity was increased compared to E6. Statistical significance was determined by an unpaired two-sided t-test. Data is shown as the mean of three replicates ± standard error and all individual data-points. **c** Genomic context of two genes; g7793 (encoding a putative galactose oxidase precursor) and g7795 (encoding a putative subtilisin-like protease PR1C). Their surrounding regions have been extensively remodeled in NE (upper panel) by the presence of two copies of the hAT-Restless TEs and a nearby breakpoint compared to their syntenic regions in E6 (lower panel), where orthologous genes are shown in the same color. **d** Possible effect of an integration of a TE on the organization of the putative destruxin A cluster. The destruxin synthase gene, as the core biosynthetic gene (dark green), and several additional biosynthetic genes (light green) are present in both E6 and NE. The additional biosynthetic genes g11098, g11099, and g11100 have been lost from the cluster (and from the genome) in NE, possibly due to the insertion of a copy of a Helitron TE.

overall divergence among the expanded TEs (median within-family divergence <0.05; for 11 of 18 TEs with copies outside the *Starship*, the median divergence is 0). Accordingly, we cannot conclude that TE copy age differs between s00261 and the rest of the genome, and our data do not resolve whether s00261 is a source or a sink for these expansions. Next, we examined the fraction of TEs that were expanded (≥1 additional copy elsewhere in the genome with >99% identity and >99% coverage), restricting to TEs >1000 bp. A significantly higher fraction of TEs on *Starship* s00261 were expanded (20/26) than in the rest of the genome (488/1001) (Fisher's exact test, *p* = 0.0049; Supplementary Fig. 7b). Notably, 60.8% of expanded TEs outside the *Starship* (297/488) had a high-identity copy inside s00261, indicating that more than half of all expanded TEs within the NE strain are connected to the *Starship* s00261. The degree of expansion, quantified as the number of near-identical copies per TE, was 8.51× higher for TEs on s00261 compared with TEs elsewhere that lacked a *Starship* copy (negative binomial Generalized Linear Model (GLM) with LRT: rate ratio = 8.51, 95% CI 6.38–11.59; *p* < 1 × 10⁻¹⁶; Supplementary Fig. 7c). Genome-wide mapping of expanded TEs in the NE genome revealed no additional clusters independent of s00261 (i.e., whose TEs lacked copies inside the *Starship*). Taken together - the evidence for horizontal transfer of both the TEs and *Starship* s00261, the higher fraction of expanded TEs on the *Starship*, and their markedly elevated copy numbers - these results indicate that s00261 likely introduced the TEs and was the source of the subsequent TE expansion in the NE genome, rather than the TEs having been horizontally transferred and expanded independently of the *Starship*.

**Structural reorganization of the *M. anisopliae* strain NE genome is associated with reduced pathogenicity**

To assess the phenotypic impact of the TE burst and genome reorganization, we compared the pathogenicity of *M. anisopliae* strain NE to strain E6 in Asian blue tick *Rhipicephalus microplus*, an economically important tick of mammal livestock that can be controlled through applications of *M. anisopliae* as a biocontrol agent. Strain NE exhibited significantly lower pathogenicity than E6 (Fig. 4a), being not different from the control treatment, indicating that, unlike E6, NE is unable to kill *R. microplus*. The reduced pathogenicity in NE compared to E6 was not restricted to *R. microplus*. In a previous study *M. anisopliae* strain NE was less infective than the E6 strain against the cotton stainer bug *Dysdercus peruvianus*, an insect pest causing heavy losses in cotton plantations[84]. NE had also a reduced sporulation capacity compared to E6[84]. Taken together, these earlier and our results here indicate that NE has decreased pathogenicity. Therefore, we tested whether the reduced pathogenicity of NE could have been a consequence of the genome reshuffling. During early infection, enzymatic activity in the fungal secretome plays a critical role in cuticle degradation[85–88] and thus we measured the 48 h differential mycelial secretome, induced in vitro by liquid culture supplemented with tick cuticle. In the secretomes, we determined the activity of six infection-related enzymes. Five of the six tested enzymes showed reduced activity in NE, while PR2 activity was increased, suggesting that strain-specific differences potentially emerge early in infection and may contribute to the observed variation in pathogenicity (Fig. 4b). Proteomic analysis of the secretome further revealed major differences in its composition: 228 proteins were more abundant in E6, 16 in NE, and 51 showed similar levels between the strains (Fig. 4c; Supplementary Data 5), highlighting a pronounced divergence in secretome profiles induced by host components.

To test whether the structural variation of the genome might have caused these differences, we then assessed whether genes encoding the secreted proteins were located near structural variations (expanded TEs, breakpoints, or *Starship* elements). Although a greater proportion of differentially expressed genes (53 of 244) were located

within 10 kb of such variations compared to similarly expressed ones (5 of 51), this trend was not statistically significant (Fisher's exact test, $p = 0.07763$). For some important individual genes, however, there appeared to be a correlation. Of particular interest is a subtilisin-like protease, PR1C, a key pathogenicity factor potentially contributing to host specificity[87,89,90]. In strain E6, the gene encoding PR1C is located on accessory chromosome chrE (geneID: g11457), whereas in NE, it has been relocated to chr4, near a copy of the hAT-Restless TE, which had recently increased in copy number, and a breakpoint where regions syntenic to chrE and chr5 in strain E6 have been fused (Fig. 4c, geneID: g7795). PR1C showed reduced expression in the secretome of NE compared to E6, which is also reflected by the lower detected enzyme activity for PR1 (Fig. 4b). Furthermore, a neighboring gene (E6 geneID: g11461) encoding a putative galactose oxidase precursor, located near the PR1C-encoding gene in E6 was also relocated by the structural variation in NE (NE geneID: g7793) and similarly showed reduced protein expression in the NE compared to the E6 secretome (Fig. 4c). In addition, a putative biosynthetic cluster for the insecticidal metabolite destruxin[91] was modified and several additional biosynthetic genes have been lost from the cluster and were also not present in the rest of the genome, possibly due to an insertion of a copy of the Helitron TE family in the NE strain (Fig. 4d). Taken together, these findings suggest that structural variation in NE may have altered the local genomic environment of many genes, leading to altered expression of key secreted proteins or insecticidal compounds and potentially explaining the strain's loss of virulence and pathogenicity.

## TEs are frequently found on *Starships* and often have additional copies in the rest of the genome

We next broadened our analysis to other fungal taxa and investigated whether TEs are generally prevalent on *Starships* and, if so, whether these *Starship*-associated TEs also have copies in the rest of the genome. To this end, we analyzed 522 published and unique *Starships* available in the STARBASE repository[92]. We included the 39 *Starships* identified in the *Metarhizium* genus in this study, thereby expanding the dataset to *Starships* from a total of 164 distinct species, and annotated all known predicted TE families using EDTA[93]. The majority of *Starships* contained known TEs, which in some cases accounted for more than 50% of the sequence, with a maximum of 71.8% TE coverage observed in *Starship* SSA002849 from *Aspergillus chevalieri* (Supplementary Data 4). In addition to *Aspergillus*, high TE contents (>30%) were also observed in *Starships* from the fungal genera *Histoplasma*, *Alternaria*, *Metarhizium*, *Lassallia*, and *Pyricularia* (Supplementary Data 4). To test whether there is an association between TE content and *Starships* we analyzed whether TE content on *Starships* differs from that outside *Starships*. In all 193 genomes in which *Starships* had been described, we annotated TEs using Earl Grey[94]. TE content was compared using two approaches: (i) assuming random TE placement outside the *Starship*, and (ii) assuming random placement of the *Starship* itself. The first assesses whether *Starship* TE content is higher or lower than the genome-wide average outside the *Starship*; the second asks whether it is more extreme than *Starship*-sized regions elsewhere in the genome. We found that 26 *Starships* differed significantly from the average TE content of their host genome (23 *Starships* had a higher TE content than the host genome and three a lower TE content; Wilcoxon test on randomly selected 2 kb windows inside vs. outside *Starships*; False Discovery Rate (FDR = 10%)), and six had significantly higher TE content than *Starship*-sized regions outside the *Starship* (permutation test, 100,000 permutations; FDR 10%) (Fig. 5a, b; Supplementary Data 4). We therefore conclude that TEs are frequently present on *Starships*, and in some cases, *Starship* TE content is significantly different, most often higher, than that of the surrounding genome.

We then assessed whether TEs identified on *Starships* had either highly similar copies ( >99% sequence identity and >99% coverage), or

perfect copies (100% sequence identity and 100% coverage) elsewhere in the genome. Of the 363 *Starships* for which this analysis was possible, 275 contained at least one TE. Of the 275 *Starships*, 35% (95) and 24% (67) had a highly similar or a perfect copy of a TE elsewhere in the genome, respectively. Of the latter, 35 consisted of TEs longer than 1000 bp (Supplementary Data 4). A higher proportion of *Starships* containing TEs with perfect copies was observed in species of *Aspergillus*, *Metarhizium*, *Alternaria*, *Colletotrichum*, and particularly *Pyricularia*, where all five of the six described *Starships* from two different genomes contained at least one TE with a perfect copy elsewhere in the genome (Fig. 5c; Supplementary Data 4). Hence, we conclude that there has been a recent exchange of TEs between *Starships* and their surrounding genomes. Either the TEs originated in the genome and transposed into the *Starship*, or vice versa.

Many *Starships* contained more than one TE with a perfect copy elsewhere in the genome in which they reside in. The maximum number of different TEs on a single *Starship* with perfect copies elsewhere in the genome was 43 in the above-described s00261 in *M. anisopliae* strain NE (Fig. 5d), 28 in *Starship* SSA003960 of *Pyricularia oryzae* (Fig. 5e), 13 in s00100 of *M. brunneum* (Supplementary Fig. 8c), and six (out of a total of 15 TEs) in SSA003873 of *Pyrenophora tritici-repentis* (Fig. 5f; Supplementary Data 4). Again, the number of very similar (>99% sequence identity with >99% coverage) or identical (100% sequence identity and coverage) TE copies elsewhere in the genome can be very high. For example, in *Starship* SSA003960 from *P. oryzae*, 28 TEs that have at least one perfect copy outside the *Starship* have a total of 277 perfect copies elsewhere in the genome, along with 1031 very similar copies. See also additional examples in Supplementary Fig. 8a–d.

## *Starships* from different species contain identical TE copies

We next asked whether different *Starships* - either from the same or different species - can share identical TEs. To test this, we searched for perfect copies of TEs (100% sequence identity and 100% coverage) located on different *Starships*. Among the 408 *Starships* with available species-level information, 35% (141) harbored at least one TE that had a perfect copy on another *Starship* in the same or in a different species. This phenomenon was particularly common in *Starships* from the genera *Aspergillus*, *Fusarium*, *Metarhizium*, *Penicillium*, *Alternaria*, and *Pyricularia* (Supplementary Fig. 9a; Supplementary Data 6). Remarkably, 12% (49) of all *Starships* contained at least one TE with a perfect copy on a *Starship* from a different species, of which in six *Starships* at least one shared identical TE was longer than 1000 bp (Supplementary Data 6). Tad1 LINE elements appear especially prone to being shared between different *Starships*, both within and across species (Supplementary Fig. 9b). We found that identical TEs are almost exclusively shared between *Starships* from species within the same genus, and that some genera are more prone to this sharing than others. In the genus *Aspergillus*, which currently has the highest number of annotated *Starships*, identical TEs are shared between *Starships* from eight different species, with the highest number of shared elements found between *Aspergillus flavus* or *Aspergillus oryzae* and *Aspergillus sojae* (Supplementary Fig. 9c). In *Metarhizium*, identical TEs are shared among five species, with the most frequent sharing observed between *M. anisopliae* and *M. brunneum* (Supplementary Fig. 9d).

## *Starships* contain TEs that show a high likelihood of past horizontal transfer

We note that identical or near-identical TEs on *Starships* from different species are not, by themselves, proof of HTT; as vertical inheritance from a common ancestor is also possible. We thus assessed HTT following Wallau et al.[32] using three lines of evidence: (i) host-TE phylogenetic incongruence, (ii) patchy TE distribution across species, and (iii) high interspecific TE similarity. Because we observed many near-identical or identical *Starship*-associated TEs in *Metarhizium* and

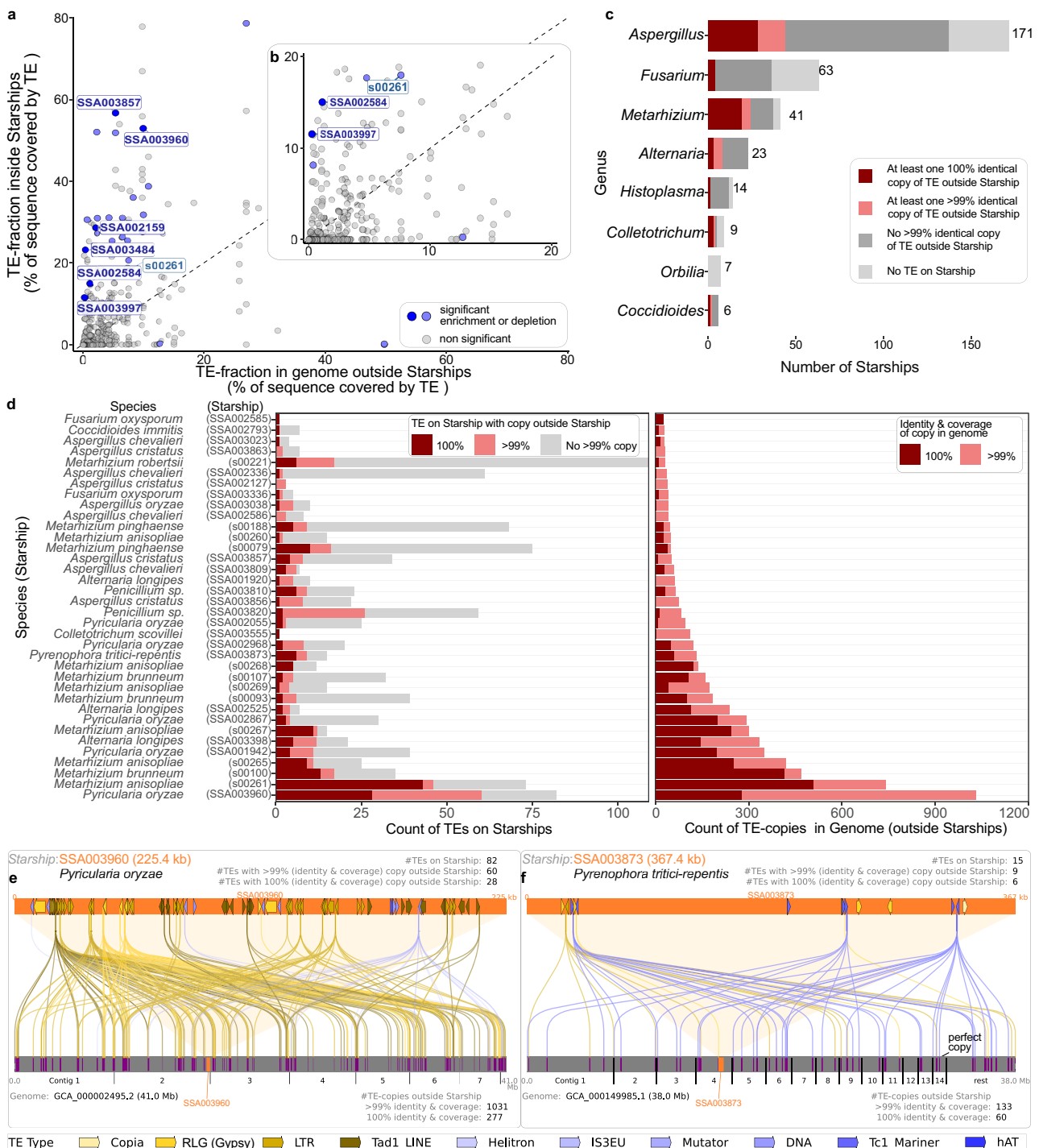

**Fig. 5 | The majority of *Starships* contain TEs, many of which have multiple identical copies elsewhere in the genome. a** Comparison of TE content between a *Starship* and the genome outside the *Starship*. Significant differences (at 10% FDR) are shown for a Wilcoxon test on TE coverage in randomly chosen 2 kb windows inside vs. outside *Starships* (light blue), and for a permutation test based on randomly placing *Starship*-sized regions within the TE-annotated genome (dark blue). Permutation-test significant *Starship*-IDs and *Starship* s00261 are indicated. **b** Zoom-in for fractions <20%. **c** Counts of *Starships*, for which genomic data was available, categorized by TE copies: *Starships* without TEs (light gray), *Starships* containing TEs but none having a highly similar copy outside the *Starship* (dark gray), and *Starships* containing TEs with at least one highly similar copy outside the *Starship* (light red), and *Starships* with at least one identical genomic TE copy outside the *Starship* (dark red). *Aspergillus*, *Metarhizium*, and *Pyricularia*, show a

high fraction of *Starships* with TEs with identical copies outside the *Starship*. Only genera with at least five *Starships* and available genomic data are shown. **d** Detailed analysis of individual *Starships* with at least 25 TE copies outside the *Starship*. Left panel: Number of TEs on *Starships*, distinguishing between those TEs with at least one highly similar (light red) or identical (dark red) copy outside the *Starship*. Many *Starships* harbor multiple TEs with highly similar or identical genomic copies. Right panel: Count of TE copies outside these *Starships*, with several *Starships* showing high numbers of very similar or identical copies elsewhere in the genome. **e, f** Graphical representations of TEs on two example *Starships* (orange, top) and their perfect genomic copies outside the *Starship* (purple markers on continuous gray genome bars, bottom): SSA003960 in *Pyricularia oryzae* (**e**) and SSA003873 in *Pyrenophora tritici-repentis* (**f**).

*Aspergillus*, we focused on these genera. Species trees were inferred from single-copy BUSCO orthologs for these genera (*Metarhizium* spp.: 347 genes; Supplementary Fig. 2a, b; *Aspergillus* spp.: 411 genes; Supplementary Fig. 10). For TEs > 1 kb located on a *Starship*, we searched all isolates using relaxed thresholds (≥80% identity and ≥80% coverage) to detect presence/absence. To test phylogenetic incongruence, alignments of TE sequences meeting these thresholds were evaluated against the single-copy BUSCO species trees using the AU test[81]; rejection indicated distinct TE versus host histories, which could indicate HTT. Finally, to evaluate "high similarity," we restricted comparisons to species (or strain) pairs with genome-wide synonymous substitutions per synonymous site (dS) $\geq 0.02$ (to ensure stable distance estimates and stable detection of reduction from this rate). Within those pairs, we considered HTT to be highly likely when the entire TE was identical between species, or - if not - when the largest open reading frame (ORF) within the TE had a much lower (≤10%) dS (termed *dS(TE)*) of the species-pair dS (termed *dS(Species)*).

We found that in *Aspergillus*, 15% (25/171) of *Starships* contain at least one TE with a high likelihood of past horizontal transfer (as defined by the criteria of Wallau et al.[32]); in *Metarhizium*, the fraction is 33% (13/39) (Fig. 6a). At the level of individual *Starships*, several show a high proportion of such TEs (Fig. 6b). For example, all eight TEs (>1 kb) in SSA003776 meet these criteria, with very closely related TE sequences present in this *Starship* from *Aspergillus luchuensis* and the distantly related *Aspergillus niger* (Supplementary Fig. 11c, e). Additional cases include SSA002946 in *A. sojae* and s00100 in *M. brunneum*, where most TEs show a high likelihood of horizontal transfer (Fig. 6c–f). These TEs are either identical or have very low *dS(TE)* in more distant species, significantly reject the species phylogeny in AU tests, and exhibit patchy distributions across congeners. Alignments of these TE sequences show very few SNPs; notably, a > 4 kb TE (ID: SSA002946_TE_12) that is identical between SSA002946 and a copy in *A. flavus*/*A. oryzae* (Fig. 6g). Together, these lines of evidence suggest that horizontal transfer occurred for multiple *Starship*-associated TEs and, in some cases, this involved a high proportion of all TEs (>1 kb) present on the *Starship*.

We next asked more generally whether *Starships* act as sources or sinks for horizontally transferred TEs. In a "source" scenario, a horizontally transferred *Starship* brings TEs into a new host, after which those TEs expand. Alternatively, the *Starship* itself transfers horizontally, while the TEs also horizontally transfer (independently of the *Starship*) and only later transpose onto the *Starship*. This is fundamentally an age question: if a TE copy on the *Starship* is older than copies elsewhere in the genome, that suggests it arrived with the *Starship*. To answer this question, we used the analysis of *Starship* s00261 (described above) as a template for the analysis of TEs > 1 kb across 25 *Starships* with large TE expansions. For each TE, we gathered all copies inside and outside the *Starship* that showed at least 99% sequence identity and coverage, built a family consensus, and used each copy's divergence from consensus as a proxy for age. We found that in 10 *Starships* (for a total of 17 TEs), the *Starship* copy was the most divergent (oldest) copy (Supplementary Fig. 12a), consistent with the *Starship* delivering the expanding TE. Conversely, we also found expanded TE copies on *Starships* that were younger than the typical copies outside the *Starship*, implying expansions that did not originate on the *Starship*. Further signals that could indicate whether a *Starship* is the source or the sink of expanding TEs were mixed: the fraction of TEs with at least one near-identical genomic copy (>99% identity and coverage) was significantly higher for *Starships* s00261 (already presented above; Supplementary Fig. 7b) and SSA002968, but lower for s00100 (Supplementary Fig. 12b), and total copies per TE were significantly higher in six *Starships* (Supplementary Fig. 12c). Together, these lines of evidence indicate both dynamics could occur: some expanding TEs likely arrived on the *Starship* and then amplified, whereas others expanded first and only later transposed onto the

*Starship*. Ultimately, *Starships* cannot serve as vectors for HTT without the prior transposition of TEs onto the *Starship*, making the acquisition of these elements a critical prerequisite for their future horizontal transfer.

In line with this, we see evidence for both scenarios when further examining individual *Starships*. i) *Starship*-delivered TEs that expand: a Tad1 LINE retrotransposon appears to have been horizontally transferred between *M. brunneum* and *M. pinghaense* and subsequently expanded in *M. brunneum*, likely invading additional *Starships* (Supplementary Fig. 13a). ii) Actively expanding TEs that later enter *Starships*: four *Starships* from *A. oryzae* and *A. sojae* share synteny containing six TEs (Supplementary Fig. 13b); on two of these *Starships*, numerous TE copies insert at positions that disrupt synteny, consistent with TE integration after *Starship* transfer. Together, these examples support reciprocal exchange between *Starships* and the rest of the genome: *Starships* can both deliver TEs that later amplify and receive TEs that are actively expanding, facilitating horizontal TE movement within and across fungal taxa.

## Discussion

Here, we provide evidence that in *M. anisopliae*, a large number of TEs were introduced into the NE strain likely via a horizontally transferred *Starship*. These TEs most likely transposed actively afterward, dramatically increasing their copy number and ultimately contributing to genome-wide rearrangements and a loss of pathogenicity. Our broad comparative analysis of 522 *Starships* across the *Pezizomycotina* subphylum further reveals that many *Starships* carry TEs and differ from the rest of the genome in TE content. Many of the *Starship*-associated TEs have identical copies elsewhere in the genome, and critically, identical or very similar copies are also found on other *Starships*, even across species boundaries. We further show that many *Starships* in the *Metarhizium* and *Aspergillus* genera carry TEs that have most likely been horizontally transferred between species, suggesting that *Starships* are important facilitators of horizontal TE transfer. The high similarity among copies further indicates that these transfers were very recent.

The key question is whether TEs arrived on *Starships* and then expanded, or whether they were horizontally acquired by independent means and subsequently spread to *Starships*. Two lines of evidence suggest the former. First, in *Metarhizium* and *Aspergillus*, the two genera analyzed in more detail, the most parsimonious explanation for the high fraction of TEs on *Starships* that show patterns of horizontal transfer is that the TEs and the *Starships* were horizontally transferred together, rather than via two independent processes. Second, on *Starships* with many expanded TEs, some of those elements display the greatest divergence from their family consensus, consistent with *Starships* being the origin of those expansions. Nevertheless, we also find evidence for the reverse, expanding TEs being copied onto *Starships*. *Starships* therefore, appear to be involved in TE biology and transposition, and should thus be recognized as vectors for horizontal transposon transfer. They seem to be not only sources of expanded TEs, but we believe that they are also vehicles that shuttle them between species within the *Pezizomycotina*.

We found identical TEs on *Starships* isolated from different species, for some of which the TEs recently must have increased in copy number. Certain genera - *Aspergillus*, *Metarhizium*, *Penicillium*, *Pyricularia*, and *Fusarium* - appear particularly susceptible to *Starship*-mediated HTT, although this observation may be biased by non-random sampling in *Starship* detection studies. TEs are predominantly shared among species within the same genus, although in some cases, identical and near identical (four and nine, respectively belonging to TC1 Mariner, RLG ("Gypsy"), Bel Pao LTR, and Mutator superfamilies) TEs were found on *Starships* across different genera. This pattern parallels the higher frequency of *Starship* transfer among closely related species compared with more distantly related ones[68] and

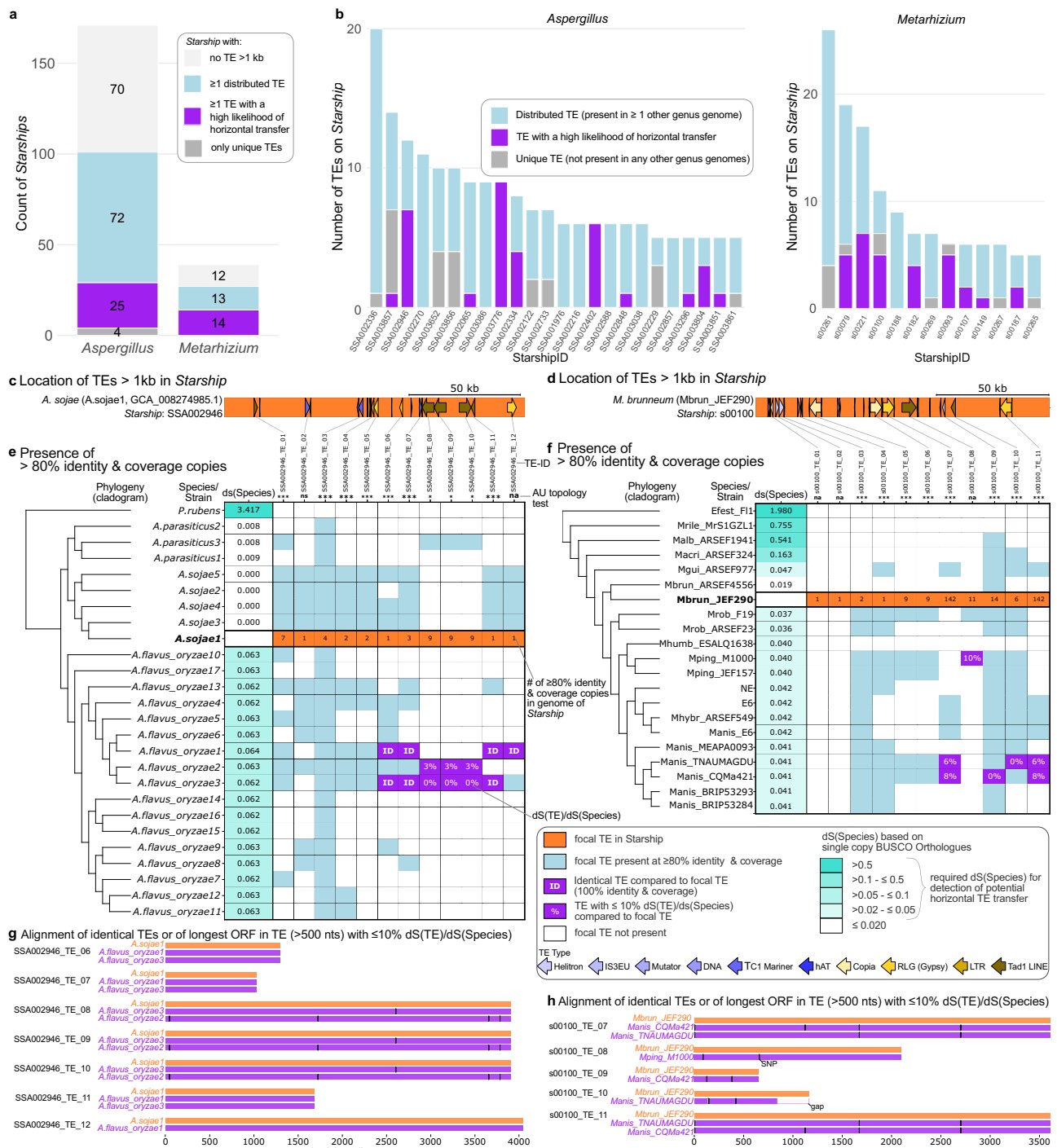

**Fig. 6 | *Starship*-associated TEs in *Aspergillus* spp. and *Metarhizium* spp. with a high likelihood of horizontal transfer. a** Overview of *Starships* and their TEs (>1 kb) in *Aspergillus* spp. and *Metarhizium* spp., indicating whether a *Starship* contains TEs with a high likelihood of horizontal transfer as defined by Wallau et al.[32]. **b** Per-*Starship* summary of TEs >1 kb. Color code for **a** and **b**: gray = unique TE (present only in the host genome at ≥80% identity & coverage), blue = distributed TE (present in ≥1 other genome), purple = distributed TE that meets the Wallau et al. criteria (patchy distribution, host-TE phylogenetic incongruence, and high interspecific similarity). **c–h**: Details for two *Starships* (left: SSA002946; right: s00100). **c**, **d**: TE maps (>1 kb) showing TE types on the focal *Starship* (orange). **e**, **f**: Presence/absence matrices across congeners (subset shown for clarity in *Aspergillus*; analyses included all *Aspergillus* genomes with *Starships*). Left margins show single-copy

BUSCO gene species cladograms. *dS(Species)* is the median pairwise *dS* between single-copy BUSCO orthologs of the focal genome and each compared genome. Blue tiles indicate detection of the focal TE in other genomes at ≥80% identity and coverage. Purple tiles indicate TE copies that are either identical (ID) or have *dS(TE)* ≤10% of *dS(Species)* for pairs with *dS(Species)* ≥ 0.02. Above each TE, results of one-sided AU topology tests for congruence between the TE alignment and the species tree are shown (ns, not significant; *$p < 0.05$; **$p < 0.005$; ***$p < 0.0005$; na, not applicable due to too few distinct sequences with a minimum of ≥3) **g**, **h**: Alignments of the TE (or its longest open reading frame (ORF)), for cases meeting the Wallau et al. criteria, shown against the focal *Starship* TE copy (orange), with SNPs and gaps indicated. *Note:* Species and strain names are codified according to Supplementary Fig. 10 and Supplementary Data 8.

interestingly also mirrors a recent result reporting that most HTT events also occur between closely related fungal taxa[38]. Moreover, the latter study found that HTT is particularly common in the subphylum *Mucoromycotina*, where no *Starships* have been reported, and in three classes within *Pezizomycotina* (*Sordariomycetes*, *Dothideomycetes*, and *Leotiomycetes*)[38], the subphylum in which *Starships* were discovered. This raises the possibility that the extensive HTT observed in these *Pezizomycotina* classes is, at least in part, facilitated by *Starships*, although this remains to be tested. We note that many of the TEs shared between *Starships* and their host genomes, as well as between *Starships* in different species, are likely fragments due to their short length (>100 bp). However, we observed similar results for longer TEs (>1000 bp), more likely representing full-length TEs, which supports our conclusion that *Starships* are facilitators of HTT. We also note that by using a very conservative threshold of >99% identity for individual TE elements, we have likely missed older *Starship*-mediated HTT events, as well as newer ones that are affected by an increased mutation rate due to RIP. Consequently, the contribution of *Starships* to the spread of TEs between species may be even more widespread than our analysis indicates.

We found that in 26 *Starships*, TE content deviated significantly in their TE content from the rest of the genome, assuming a random TE distribution (23 higher, 3 lower). In six *Starships*, TE content was significantly higher than expected from the empirical genomic TE distribution - meaning they are significantly higher in TE content than *Starship*-length regions within these genomes. This indicates an association between TEs and *Starships*. An explanation for the enrichment of *Starships* with TEs could be that TEs may preferentially transpose into *Starships*, although the underlying mechanism remains unclear. Some *Starships* are known to integrate into AT-rich regions, 5S rDNA loci, or even directly into other TEs[80], which are also known as preferential insertion sites for TEs. This may result in *Starships* and active TEs occurring in close proximity to each other. In addition, TE insertions also co-localize with open chromatin[95], hence if the chromatin state of *Starships* were open, then they might indeed be some of the only regions where active TEs are able to insert. But still, the apparent tendency of TEs to target (active) *Starships* remains puzzling. *Starships* are frequently horizontally transferred; for example, in the genus *Paecilomyces*, between 18–27% of *Starships* show evidence of horizontal transfer under natural conditions[68]. This high frequency of horizontal transfer renders *Starships* conceptually similar to conjugative plasmids in bacteria for horizontal transfer of sequences[65]. It might therefore be possible that active *Starships* could be preferentially targeted by TEs to increase their chances of horizontal transmission. A precedent for such a mechanism exists in bacteria: the transposon Tn7 is known to specifically insert into conjugative plasmids, likely recognizing them via features associated with lagging-strand DNA replication[96]. This strategy enhances the probability of horizontal transfer while minimizing the potential costs of transposition[96]. Given that *Starships* may be associated with circular DNA intermediates, which would be unique and recognizable[55], they could similarly provide an opportunity for TE mobility and spread. While this would represent an intriguing parallel to conjugative plasmids, the high diversity of expanded TEs (from both major TE classes) observed within *Starships* argues against a single, conserved mechanism. Irrespective of the mechanism, it has been proposed that most actively transposing TEs might enter the genome via HTT[26,28,97]. This is because defense mechanisms, such as RIP, would require time to recognize and deactivate newly introduced TEs. Hence, the horizontal transfer of TEs might be an integral part of TE life cycles, and the possibility that TEs might exploit *Starships* for horizontal transfer remains an exciting avenue for future research.

We find that expanded TEs have led to massive structural reorganization of the genome in *M. anisopliae* strain NE. It is striking that only DNA transposons are involved in the structural variation, despite the expansion of several larger (>1000 bp) retrotransposons in the same strain. Two of these DNA transposons, Mutator and hAT-Restless elements, transpose by generating double-strand breaks (DSBs) and form hairpin intermediates either in the flanking regions or within the transposon itself, through a mechanism resembling V(D)J recombination by binding to terminal inverted repeats[98,99]. These DSBs can be repaired replicatively via homologous recombination (HR) using a homologous chromosome, sister chromatid, or another genomic copy as a template, potentially leading to sequence homogenization[100,101], or through non-replicative repair via non-homologous end joining (NHEJ)[102]. The third TE family associated with breakpoints in NE is a Helitron which replicates via a unique rolling-circle mechanism (peel and paste mechanism) and is capable of capturing and mobilizing host genes[103,104]. Widespread across eukaryotes, Helitrons have significantly influenced genome evolution, with horizontal transfer events contributing to genomic changes in *Xenopus laevis* and bats[105,106]. Although Helitrons can capture adjacent sequences, these smaller elements rarely exceed 10 kb and typically carry only gene fragments rather than full-length coding sequences[103,107,108], and therefore this alone cannot account for the widespread chromosomal reorganization we observed. In conclusion, the structural variation observed in *M. anisopliae* strain NE appears to result from a combination of homologous recombination between identical TE sequences and the direct effects of transposition. Mutator and hAT-Restless elements induce DSBs, while Helitrons may introduce single-strand breaks, both increasing the likelihood of recombination. The abundance of identical, likely active TEs may fuel a self-reinforcing, runaway process. However, about half of the breakpoints lack identifiable TEs, and the low sequence conservation in these regions suggests the involvement of additional, as yet unknown, mechanisms. Overall, the extensive structural changes observed in *M. anisopliae* strain NE are at least partially driven by the expansion and activity of these likely *Starship*-derived DNA transposons.

The observed chromosomal rearrangements could have a major impact on fungal speciation, in this particular case, in the taxon *M. anisopliae*. The burst of TE activity and the resulting extensive chromosomal reorganization in the strain NE, compared to the more conserved E6 strain, may lead to rapid genetic differentiation and reproductive isolation, as previously hypothesized[109]. Such an HTT, followed by TE proliferation that induced genome rearrangements and in turn contributed to (partial) reproductive isolation, has already been reported in *S. paradoxus*[33]. Although meiotic recombination has not yet been documented in *M. anisopliae*, it seems highly unlikely that recombination would be successful between the extensively reshuffled genome of strain NE (or its descendants) and the more ancestral genome of E6. In diploid heterozygotes, such divergence could disrupt chromosomal pairing, crossing over, or segregation during meiosis. As a result, the offspring would likely be inviable (or completely sterile) or suffer from drastically reduced fertility or fitness (reviewed in ref. 110), and hence *M. anisopliae* strain NE is possibly now reproductively isolated – and could be considered an incipient species. Additionally, SVs may alter the epigenome, gene regulatory networks, or chromatin organization within the nucleus[111]. Such changes could explain the observed differences in the secretome between strains NE and E6, which cannot be entirely attributed to the vicinity of the corresponding genes to TE insertions or breakpoint locations alone. Regardless of its impact on potential speciation, the extreme structural genome reorganization observed in the NE strain is correlated with changes in pathogenicity and life history strategy. Our results indicate that most proteins previously linked to pathogenicity and associated with TEs, breakpoints, or *Starships* (proteases, chitinases, oxidoreductases, and other effector proteins) were down-regulated in the NE strain. This provides another compelling example of structural variation as a major driver of genomic and phenotypic diversity, adding to the known cases reported among pathogenic fungi[112].

In conclusion, we here show that *Starships* could be vehicles for the horizontal transfer of TEs, which can subsequently expand and exert substantial effects on recipient organisms. As such, *Starships* could be important vectors for the horizontal spread of active TEs.

# Methods

## Strains and Culture

*M. anisopliae* strain E6 and NE were isolated from *Deois flavopicta* and *Mahanarva posticata* insects, respectively. Both strains were isolated from Brazil: E6 from Espírito Santo state and NE from Pernambuco state.

The strains E6 and NE were cultivated for conidia and mycelia production as previously described[113]. Briefly, spores ($10^6$ mL$^{-1}$) were added to propylene bags containing 100 g of rice and 30 mL of 0.5% peptone, and left at 28 °C for 14 days to promote conidia production. The dry spores were collected for further suspension production. Then, spores ($10^7$ mL$^{-1}$) were inoculated in 70 mL of basal medium (0.6% NaNO$_3$, 0.2% glucose, 0.2% peptone, and 0.05% yeast extract w/v) encompassing cuticles of *R. microplus* (0.7%) and cholesteryl stearate (0.05%), for triggering the infection machinery[90]. The flasks were incubated at 28 °C with shaking (150 rpm) for 48 h. After growth, 0.25% (v/v) Triton X-100 was added and mixed to extract enzymes and proteins from the mycelia's external surface. Afterwards, the mycelia were harvested by filtration through a Whatman no. one filter paper. These filtrates containing secreted proteins were frozen and later used for enzymatic assays.

## DNA extraction

For DNA extraction, cells were propagated in liquid culture using SBD medium with a final glucose concentration of 8% at 28 °C and 200 rpm. Cells were harvested by centrifugation (3200 x *g*, 20 min at RT) and washed twice with H$_2$O before the supernatant was removed and the DNA was purified using a modified CTAB protocol with two phenol: chloroform extractions[114].

## Sequencing and pulsed-field gel electrophoresis

DNA was sequenced at BGI Genomics, Hong Kong, using the PacBio Revio system using the SMRTbell® Prep Kit 3.0 (PacBio, PN 102-182-700) for library preparation and resulting in Hifi reads. Pulsed-field gel electrophoresis was conducted, and gel bands were extracted and sequenced as described elsewhere[48].

## Enzymatic activities

The enzymatic assays were carried out as previously described[115]. All assays were performed in three replicates using culture supernatant prepared as described above. All enzymatic activities were expressed as specific activity (U/mg of protein) based on total protein content measured using the bicinchoninic acid (BCA) protein assay (Pierce, Rockford, USA). The statistical analysis was conducted by an unpaired t-test using GraphPad software.

For lipase activity, solution 1 (ρ-nitrophenyl palmitate (ρNPP) (Sigma, USA) 3 mg/mL in isopropanol) was added in droplets to solution 2 (Tris-HCl 50 mM pH 8.0, 0.11% gum arabic, 0.44% Triton X-100) in a 1:9 (v/v) proportion under strong magnetic agitation, resulting in the substrate solution. Subsequently, the supernatants (10 µL) were mixed with the substrate solution (90 µL) and read immediately at 410 nm (time zero). The mixture was incubated at 37 °C and read again after 30 min in the SpectraMax spectrophotometer (Molecular Devices, USA). As a control, a buffer was added instead of the secreted extract. Results were obtained in µmol of ρNPP per hour.

Protease activity was evaluated using 2 mM of subtilisin (Pr1) substrate (N-suc-ala-ala-pro-phe-ρNA) and trypsin (Pr2) substrate (Bz phe-val-arg-ρNA) in 100% DMSO. Approximately 2 µg of protein from each supernatant was added to a 50 mM Tris-HCl pH 8.0 buffer to complete 100 µL. Kinetic assays were monitored at 37 °C for 30 min in

a SpectraMax. One protease unit (U) was defined as the amount of enzyme that produces one ρmol of ρ-nitroaniline per hour, in the assay conditions described.

For catalase activity, hydrogen peroxide was used as substrate, and phosphate buffer was added along with the substrate at 10 mM to 25 µL sample aliquots. The reduction in hydrogen peroxide concentration was tracked by measuring the decrease in absorbance at 240 nm over a period of three minutes (E = 39.4 mM cm$^{-1}$).

For phospholipase C (PLC) activity, ρ-nitrophenylphosphorylcholine (ρNPPC) was used, which is a chromogenic substrate selectively hydrolyzed by PLC. The ρNPPC was prepared at a 20 mM concentration in a 50 mM Tris-HCl pH 8.0 and 60% sorbitol buffer. Samples (10 µL) were mixed with 90 µL of substrate solution and incubated at 37 °C for 1 hour. Control was made with a buffer instead sample. After this time, the assay was read at 410 nm in a SpectraMax spectrophotometer. One unit of PLC was defined as described above for the lipolytic activity unit.

For phosphatase activity, the rate of ρ-nitrophenol (ρ-NP) production was assessed according to[116]. Samples were incubated for 1 hour at room temperature in 0.2 mL of reaction mixture (116.0 mM NaCl, 5.4 mM KCl, 30.0 mM Hepes-Tris buffer, pH 7.0, and 5.0 mM ρ-NPP). Enzymatic activity was triggered by adding the samples and halted with 0.2 mL of 20% trichloroacetic acid. Then, an aliquot of 0.1 mL of the supernatant was added to a 96-well plate, along with 0.1 mL of NaOH. Readings were made at 405 nm using the SpectraMax spectrophotometer. The enzymatic activity was then calculated by subtracting the non-specific ρ-NPP hydrolysis, measured for the control (made with buffer), and the concentration of ρ-NP was estimated using a standard calibration curve.

## Bioassays

For the bioassays, engorged *R. microplus* females from infested bovines were disinfected using hypochlorite 2.5% for 2 s, washed with sterile saline and sterile distilled water[117]. Then, ticks were completely immersed in a $10^8$ conidia mL$^{-1}$ suspension of E6 or NE strains for 15 s. After exposure, ticks were sorted into three independent groups of 10 individuals in Petri dishes and kept in a humid chamber (>90% relative humidity) at 28 °C. They were inspected to check for the surviving animals every day. Controls were immersed in sterile distilled water instead of spore suspensions, under the same conditions described here.

## Proteome analysis

**Sample preparation for mass spectrometry.** For secretomic analysis, supernatants of each strain were promptly boiled for 5 min, in order to inactivate endogenous proteases that could affect the secretome results. Then, the samples were lyophilized and stored at −80 °C until use. Then, 100 µg of proteins were resuspended in digestion buffer (8 M urea, 100 mM Tris-HCl, pH 8.5) and digested with trypsin (2 µg) (Promega, Madison, WI) for 16 h at 37 °C as previously described[90]. The reaction was then stopped by adding 5% formic acid.

After digestion, the proteins were pressure-loaded into a biphasic capillary column containing 2.5 cm ion exchange resin (Partisphere SCX) and 2 cm reverse phase resin (Acqua C18). MudPIT salt separation included twelve steps, utilizing a gradient ranging from 0 to 100% of buffer B (80% acetonitrile/0.1% formic acid)[118]. Peptide samples were then loaded onto an LTQ-XL system (Thermo Fisher, USA), following the manufacturer's protocol. Each analysis cycle consisted of one full-scan mass spectrum (300–2000 m/z) followed by five data-dependent MS/MS spectra at a 35% normalized collision energy, which was repeated continuously throughout each step of the multidimensional separation. Dynamic exclusion was enabled with a repeat count of 1, a repeat duration of 30 s, and an exclusion list size of 200, as previously described[90] in order to prevent repetitive analysis. The Xcalibur data system (Thermo, CA) was used to control both the mass spectrometer scanning and the HPLC solvent gradient functions. Mass spectrometry

analysis was performed in six different runs per condition, with each one prepared from an independent biological replicate/liquid culture.

**Protein identification, quantification, and molecular characterization of the differential secretome.** Different software programs were used to characterize the secretomes obtained from mycelia and conidia of both strains, leading to an evaluation of the molecular and functional properties of the secretomes. PatternLab V[119] was used in the identification of the proteins, providing a relation of which were exclusively identified in each condition (AAPV module) and which were differentially expressed (TFold module), along with the up- and down-regulated proteins for each growth condition. Protein identification was based on the *M. anisopliae* strains E6 and NE sequenced genomes described here. Redundant protein sequences of the merged genomes used as a database were removed with the SeqKit2 package within the R environment, using the seqkit function[120]. The parameters used were: proteins detected in at least four out of six replicates per condition, t-test with a p-value of 0.005, and BH (Benjamini-Hochberg) q-value of 0.05 (5% FDR). The BlastP suite (https://blast.ncbi.nlm.nih.gov/Blast.cgi) was used in order to further verify every detected protein sequence, check hypothetical proteins, and re-annotate those with enough corresponding matches or identify conserved domains.

The STRING Platform (https://string-db.org), providing information on functional protein association networks, was used to categorize the differentially expressed proteins (DEPs) (*p*-values < 0.05) and Gene Ontology (GO) terms related to biological processes (BPs) and molecular functions (MFs).

The search for conserved secretion signal prediction was performed using four online programs: Phobius (https://phobius.sbc.su.se/), PrediSi (http://www.predisi.de/), SignalP 6.0 (https://services.healthtech.dtu.dk/services/SignalP-6.0/), and TargetP 2.0 (https://services.healthtech.dtu.dk/services/TargetP-2.0/). A protein was considered to have a positive prediction for a secretion signal if at least two out of the four programs yielded a positive result.

## Bioinformatic analysis

**Genome Assembly.** Assemblies were generated from PacBio Hifi Reads using Canu (version 2.2)[121]. Assemblies were visually inspected using Tapestry[122], and contigs that showed less than 20% or more than 300% of the average genome-wide coverage - as calculated by mosdepth (version 0.3.4)[123] on the minimap2 (version 2.26-r1175)[124] aligned PacBio reads - were removed, because these were considered to be mitochondrial genome, spurious contigs, or contamination. The resulting assemblies were scaffolded using the PacBio reads by ntLink (version: v1.3.11)[125], which led to seven contigs and three scaffolds in strain NE, and two contigs to one scaffold in strain E6, leaving all other contigs unmodified. These final assemblies were again visually inspected by Tapestry, and telomeric sequences (TTAGGG) were detected. The general assembly statistics calculated using Tapestry[122], are provided in Supplementary Data 1, which also includes the total number of genes and transcripts and the fraction these cover in the sequence of each contig/chromosome. For E6, all scaffolds have telomeric repeats at both ends, therefore representing chromosomes, while in NE, three scaffolds have telomeres at both ends, three have telomeres only on one end, and two have no telomeric sequences. The mitochondrial genomes were identified by MitoHifi (version: v3.2.2)[126].

Small (accessory) chromosomes were identified by extracting DNA from chromosomal bands separated by pulsed-field gel electrophoresis as described in ref. 48. The extracted DNA was then sequenced using a low-input DNA library preparation protocol at the Max Planck Sequencing Center in Cologne on an Illumina Next-Seq2000 platform. Reads were trimmed using Trimmomatic[127], and quality was inspected using FastQC (version v0.12.1)[128] and mapped to the respective genome assemblies using Bowtie2 (version 2.5.1)[129]. Duplicates were marked, and read groups were added using Picard.

Sequencing coverage in 50-kb windows was calculated using mosdepth (version 0.3.4)[123].

**Genome annotation.** The genome assemblies were annotated for TEs, genes, secondary metabolite cluster candidates, functional gene annotation, BUSCO completeness (against the *ascomycota_odb10* dataset), tRNA repertoire, gene-wise relative synonymous codon usage (RSCU), potentially secreted proteins, effector candidates, and CAZymes. The annotation was performed using a combination of tools: EDTA (version 2.2.2)[93] (for the *Starships*) or Earl Grey (version 5.1.1)[94] (for fungal genomes including the *Starships*) using the dfam version 3.8[94], BRAKER3 (version 3.0.8)[130] with fungal protein information from the OrthoDB11 database as evidence, antiSMASH (version 7.1.0)[131], BUSCO (version 5.7.1)[132], eggNOG-mapper (version 2.1.12)[133], and InterProScan[134] using the interproscan-5.68-100.0 database. tRNA genes were identified using tRNAscan-SE (version 2.0.12)[135], and functional gene annotation was further supported by BLAST[136] searches against the SwissProt database restricted to fungi. Codon usage and RSCU values were calculated using bioKIT (version 1.1.3)[137], and CAZymes were annotated with dbCAN[138]. Functional annotations were collated and visualized using Blast2GO (version 6.0.3)[139]. *Starships* were identified among 19 *Metarhizium* spp. genomes and annotated using Starfish[80]. *Starships* were visually confirmed and only *Starships* with at least 50,000 bp total flank-alignment were kept. Phylogenetic trees were constructed on MAFFT alignments using IQ-TREE2[140] with 1000 bootstrapping and automatic model detection. Phylogenetic trees were visualized using iTOL[141]. Please note that two different software versions were used to annotate TEs: EDTA (v2.2.2) and Earl Grey (v5.1.1). EDTA was initially used to annotate TEs within the *Starships*; however, it was later replaced by Earl Grey for all genomes, including the *Starships*. To ensure consistency, we only compared TE annotations generated by the same software (e.g., when comparing TE content inside and outside of a *Starship* in Fig. 5a and 5b, we exclusively used Earl Grey annotations).

**Variant calling and synteny analysis.** To determine the SNPs between NE and E6, the E6 PacBio reads were mapped onto the NE genome assembly minimap2 (version 2.26-r1175)[124], and subsequent SNP calling and file processing were performed with SAMtools (version 1.17)[142], BCFtools (version 1.17)[142], and BEDTools (version 2.31.0)[143].

Orthogroups were defined using OrthoFinder (version 2.5.5)[144], and synteny based on orthologous genes was determined with GENESPACE (version 1.3.1)[145]. Syntenic blocks were visually inspected, and adjacent syntenic regions with identical orientation were manually merged if they were not separated by more than 5 gene models. Breakpoint regions were defined as the boundaries between adjacent regions located on different syntenic chromosomes and/or with differing orientations. For each identified breakpoint, the region in the *M. anisopliae* NE genome plus 25 kb of upstream and downstream flanking sequence was extracted and aligned to the corresponding syntenic region in *M. anisopliae* E6 using MUMmer[146], as implemented in pyGenomeViz (version 1.5.0) (https://github.com/moshi4/pyGenomeViz/), and visualized with the same tool. Breakpoint coordinates were extracted from the MUMmer alignment output. Only those breakpoint regions where visual inspection allowed the localization of the breakpoint within a 10 bp window were retained. These 10 bp sequences, which contained the breakpoints, were subsequently aligned with each other using MAFFT (version 1.5.0), as implemented in Geneious Prime (version 2023.1.2).

***Starship* analysis.** A total of 632 *Starships* were downloaded from STARBASE[92] on 10.03.2025. After comparison and visual inspection, 618 *Starships* were determined to be likely unique. Please note: The accessions on STARBASE did change on its recent release. We have updated the accessions and removed a further 96, now considered as

duplicated, *Starships* from the analysis (final number of *Starships* included in the analysis: 522 from STARBASE, plus 39 identified in this study in *Metarhizium*. Total = 561). Hence, we provide all information on these accessions in Supplementary Data 4 and, in addition, the fasta file containing all 39 *Starships* identified in this study in Supplementary Data 7. For those *Starships* for which the genome accession was not available, the genome assembly information was retrieved from NCBI for 354 *Starships* using the genomic location IDs provided by STAR-BASE. TEs were annotated on all *Starships* (retrieved from STARBASE and those identified in this study) using EDTA, including a Helitron consensus sequence identified as having recently increased in copy number in *M. anisopliae* strain NE. This Helitron sequence was included because of the low performance of EDTA in annotating it. Only repetitive regions longer than 100 bp and assigned to known TE families were considered for further analysis; repetitive regions lacking a TE classification were excluded. FASTA sequences of annotated TEs were extracted and subjected to BLAST searches against (i) their genome of origin - where all *Starship* regions had been hard-masked - and (ii) all *Starships*. BLAST hits were filtered to retain only those with >99% sequence identity and >99% coverage of the query. These were further categorized to identify hits with 100% identity and coverage. For each *Starship*, the number of TEs with at least one qualifying BLAST hit was recorded. Since similar or identical TEs and subsequences often exist within the same *Starship*, raw counts of BLAST target hits would be inflated, BLAST target hits were merged if they overlapped >99% (of the smaller sequence) before counting. The genome accessions for publicly available genome assemblies of the *Metarhizium* species are described in Supplementary Data 8, and assemblies for published *Starships* are described in Supplementary Data 4.

**Comparison of TE content between *Starships* and host genomes.**
We annotated TEs across 193 genome accessions with Earl Grey (v5.1.1) using Dfam v3.8[94], retained only TEs assigned to known superfamilies, and excluded "Unknown" repeats. Base-pair coverage by TEs was quantified with BEDTools intersect (v2.31.1)[143] for each focal *Starship* and for the non-*Starship* portion of the same assembly. We then applied two complementary tests: (i) a window-based comparison that tiled fixed 2000-bp windows across *Starship* and non-*Starship* regions, retained full windows, randomly subsampled matched counts (every third *Starship* window), computed per-window TE coverage with BEDTools coverage, and compared groups using a two-sided Wilcoxon rank-sum test; and (ii) an empirical permutation test that shuffled same-length intervals (up to 100,000 draws) within non-*Starship* regions to generate a null distribution of TE-overlap and two-sided empirical p-values. Multiple testing was controlled at a 10% FDR.

**Horizontal transfer of TEs.** We inferred a species phylogeny for *Metarhizium* and *Aspergillus* accessions harboring *Starships* by annotating genes with BRAKER3 v3.0.8 (using OrthoDB11 fungal proteins as evidence and after having softmarked TEs)[130], inferring orthogroups with OrthoFinder (v2.5.5)[144], identifying single-copy BUSCO orthologs with BUSCO (v5.7.1)[132], randomly selecting 347 or 411 genes present in all isolates, extracting their DNA sequences (BEDTools), aligning each gene with MAFFT (v7.525)[147], concatenating alignments with AMAS (v1.0), and estimating a maximum-likelihood tree in IQ-TREE2 (v2.3.6)[140] with 1000 bootstrap replicates. To test TE phylogenies against these species tree, *Starship*-encoded TE sequences (>1 kb) were BLASTed against all genome accessions in the genus (plus outgroup); per accession we retained the best hit with ≥80% identity and ≥80% coverage, aligned each TE set with MAFFT, pruned the species tree to matching taxa, and in IQ-TREE2 fitted the best model, estimated an unconstrained ML tree and a species-constrained ML tree, then compared them using an AU test (10,000 RELL replicates)[81]; rejection of the species-constrained topology indicates phylogenetic incongruence consistent with horizontal transfer.

*dS(Species)* vs. *dS(TE)* rates. For each genome pair, coding sequences of the single-copy BUSCO orthologs were aligned with the codon-aware PRANK (v.170427)[148], pairwise dS were estimated per gene with PAML codeml (v4.10.9)[149], and the median across genes defined dS(Species). For each focal TE, the longest ORF was identified with orffinder v1.8, queried by tBLASTn against the corresponding TE hits above; the best hit per sequence was aligned codon-aware to the focal ORF with PRANK and pairwise dS was estimated with codeml to obtain *dS(TE)*, which we compared to *dS(Species)* for pairs with *dS(Species)*≥0.02.

Sequence divergence of TEs to consensus. For each *Starship*, all annotated TEs were extracted, and only TEs >1 kb that overlapped the *Starship* interval were retained; their sequences were obtained with BEDTools getfasta (v2.31.1). Each retained TE was queried against its source assembly with BLASTN, and sequences of hits meeting stringent thresholds (≥99% identity and ≥99% coverage) were extracted and aligned with MAFFT v7.525[147]. A custom Python script (see code availability) computed per-hit divergence to the majority-rule consensus and the divergence of the original *Starship* TE to that consensus, ignoring gap positions. Hits were annotated as inside or outside the *Starship* region, and for each TE with ≥1 in-*Starship* hit, summary statistics (n, min, mean, median, max) of percent divergence were calculated separately for inside vs. outside hits.

**Statistical analyses.** Statistical analyses were conducted in R using RStudio (versions 2023.12 and 2025.09). Details and results of the statistical tests referenced in the figures are provided in Supplementary Data 9. Data displayed in the figures is available in the Source Data file.

Note: While scripting for the parallelization of bioinformatics procedures was aided by Large Language Models (LLMs) (ChatGPT 4.0), the entire bioinformatics procedure, including the choice and setting of tools and commands, remained the sole responsibility of the authors.

**Reporting summary**
Further information on research design is available in the Nature Portfolio Reporting Summary linked to this article.

## Data availability
The sequencing reads, genome assemblies of *M. anisopliae* strain NE and strain E6 NCBI using the bioproject PRJNA1277033 and PRJNA1277034. The genome assemblies, as well as gene and TE annotations for *Metarhizium anisopliae* strains NE and E6, are also deposited here: https://zenodo.org/records/17724081. The raw spectra data and search results files generated in this study have been deposited in the ProteomeXchange Consortium via the PRIDE partner respository under the dataset identifier PXD067418. *Starship* accessions were obtained from Starbase (https://starbase.serve.scilifelab.se/). *Starship* accessions are listed in Supplementary Data 4. Genome accessions were obtained from NCBI and listed in Supplementary Data 8. Source data are provided with this paper.

## Code availability
The detailed scripts of the bioinformatic analysis are available here: michaelH-git/StarshipTE_*Metarhizium*_anisopliae which is also available on Zenodo (https://zenodo.org/records/18234057)[150].

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

## Acknowledgements

We particularly would like to thank Adrian Forsythe, Aaron Vogan, and Emile Gluck-Thaler for providing the extremely helpful STARBASE database, without which much of the analysis would not be possible. M.H. is grateful for the support of the ERC (Project 101219076 - Mobi-Chrom), H.G.K. is grateful for the support of the Novo Nordisk Foundation (NNF23OC0086230), L.S. is grateful for the support of CNPq (303971/2025-8) and W.O.B.S is grateful for the support of FAPERGS (25/2551-0002815-9) and CNPq (303945/2025-7).

## Author contributions

H.G.K., J.F.S.A., U.O., Y.P.C., A.L.M., M.B., L.S., W.O.B.S., and M.H. performed and analyzed the experiments. M.H. and H.G.K performed the bioinformatic analysis. M.H. wrote the original draft of the manuscript, and M.H. and W.O.B.S. supervised the work. All authors revised the manuscript.

## Funding

## Competing interests

The authors declare no competing interests.
