## [Transparent Peer Review file · Nature Communications]

Transposable elements hitchhike on *Starships* across fungal genomes

Corresponding Author: Dr Michael Habig

Version 0:

Reviewer comments:

Reviewer #1

(Remarks to the Author)

In this manuscript, Griem-Krey et al. investigate the phenomenon of horizontal transposon transfer (HTT) in fungal microbes. They find that giant Starship transposons are likely a common vector of HTT, and that the introduction of transposons into new genomes has important impacts on genome evolution and the evolution of fungal pathogenicity. Crucially, the authors enhance their genomic analyses with infection assays and secretome analyses, and identify interesting associations between observed decreases in pathogenicity and structural rearrangements induced by TE expansion. The authors then take the additional step to generalize their findings by testing for TE-enrichment in a larger database of Starship sequences. The manuscript contributes a timely and impactful story to the literature but there are a couple of opportunities for improvement. Below I outline several suggestions that I believe are necessary to strengthen and support the authors' claims of HTT and TE enrichment on Starships.

The in text citations for the supplementary figures do not match the figure legends (e.g. in text, Fig S4 actually appears to refer to Fig S5...)

Line 50: more clear if you change to "to escape a specific host genome's..."

L98: missing a space between "known 618"

L224, Fig S5E: were only protein sequences aligned to make this synteny alignment? I think it would be more insightful if the alignments were based on nucleotide sequences, that way you could account more precisely for sequences that are shared/divergent between these starships

L232: TE acronym has already been defined

L231-240: I'm confused. How many TE copies are on Starship s00261? 73? And does this include the 9 copies of the TE families mentioned earlier? I think I understand, but this is a really crucial result so please clarify as best you can

L248: why would an increased density of TEs on the Starship indicate that it is a source? I guess this assumes that the TE landscape in whatever the donor genome is has a relatively high density of these TE insertions such that they accumulated in the Starship region. Could be good to be more transparent with the logic here. Also, since genomes of *M. pinghaense* and *M. robertsii* are available, why not actually test whether the density of the TEs on the Starship matches the genomic background TE density of these other species? This would strengthen your argument.

L209-231: I don't see evidence supporting the horizontal transfer of the Starship itself, beyond the analysis looking at signatures of horizontal transfer in the TE sequences. Yet the authors refer to "horizontally acquired Starships" (L231). If you want to call the Starship horizontally transferred, it would strengthen your argument if you looked for evidence of its presence in other fungi, especially the putative donors *M. pinghaense* or *M. robertsii*, beyond the individual TE sequences located on the Starship (like, what about the captain or other genes? Do they have signatures of HGT too?).

Throughout the description of these results, it may also be helpful to mention the taxonomic scope of the databases that were searched, for example, when looking for the closest relative to the Starship-borne TEs. Did this database contain only *Metarhizium* species or a more taxonomically diverse group? How many isolates per species?

L374-376: one issue with the null hypothesis that TEs are randomly distributed across the genome is that we know that this is almost certainly never the case because TEs tend to accumulate in TE-rich compartments in many fungal genomes. The question of whether TEs are enriched in Starships or whether Starships tend to be in TE-rich compartments is a bit of a chicken and egg question. The key comparison that needs to be done in my opinion is to test whether the pattern of TE enrichment observed in Starships differs from other TE rich compartments in the genome. For example, can you group TEs into genomic regions (e.g. group all TEs within 10kb of each other into the same region), and then test for enrichment of particular TEs across all these regions? Are Starships the only such region that pop out of the analysis, or are there others as well? Determining whether Starships are somehow "special" would go a long way towards strengthening your argument

that they are the source of TEs.

L408-411, L415, L424: Comparing *A. oryzae* and *A. flavus* as different “species” is not the most convincing, as *A. oryzae* is thought by some to not be a separate species at all but a domesticated *A. flavus* lineage (the two “species” share 99.5% genome-wide nucleotide similarity). I honestly don’t think its worth highlighting, because it detracts from your argument, and I don’t think you can confidently claim that similar Starships found in these two “species” were exchanged via HTT. Is there a better example? Whatever example you do pick, please make sure the species boundaries are well supported in the literature.

Can you please include estimates of how often you observe at least one TE with a perfect copy on a Starship from a different genus? You mention some on L478-479, these would be most interesting to see. This would rise above some of the issues of identifying HTT among closely related species.

L680: the URL does not work when I copy and paste, please provide full URL

Reviewer #2

(Remarks to the Author)

Griem-Krey et al. “Transposable elements hitchhike on Starships across fungal genomes”

This manuscript reports genome assemblies and annotations for 2 strains of the insect-pathogenic fungus *Metarhizium anisopliae*, and a comparative analysis of transposable element (TE) content in these strains and related species. The central claim of the paper is that a class of newly identified large TEs called Starships that undergo horizontal transfer (HT) can carry other smaller TEs that can infect new genomes and impact genome organization.

My major concern with the manuscript is that the evidence in support of the claim that Starships are vectors for horizontal transposon transfer (HTT) is reasonably good, but not as strong as the authors assert in their manuscript. There are three well-established criteria enumerated by Wallau et al (PMID: 22798449) for showing HTT, which the authors have not fully shown to prove HTT. Additionally, in many places the authors state the conclusion for Starship-driven HTT as proven, before providing evidence in support of this conclusion or considering alternative hypotheses. For this manuscript to not mislead a general readership about the strength of the evidence for the stated claims, the evidence for Starship-driven HTT needs to be framed in terms of Wallau et al’s criteria and carefully evaluated with respect to alternative hypotheses prior to making the major conclusion, which I believe would lead to a more accurate and balanced statement of the support for the authors claims.

One specific area of concern I have related to the proof that Starships are donors of HT TEs is the statistical analysis of TEs that share high identity inside and outside Starships. If I understand the argument correctly, if Starships are the donors for TEs that undergo post-HTT expansion there should be a higher density of TEs in Starships relative to the recipient genome. The authors use Fisher’s exact test to provide evidence for this prediction. I have two issues with this analysis. First, Fisher’s exact test should be used for numbers from two categories in a 2x2 contingency test for association. The authors do not provide enough information to understand how TE density inside and outside starships fits this the requirements for this test. In my opinion, this test is inappropriate, and the appropriate test should be a proportion test. Moreover, regardless of the appropriateness of the statistical test used, the result of this hypotheses test does not definitively support Starships as donors, since the alternative hypothesis of Starships being less constrained at the nucleotide level than the host genome (and hence more likely to tolerate new TE insertions) is an equally plausible interpretation of the data.

Another specific area of concern relates to the last section of the results “Starships from different species contain identical TE copies indicating HTT between species”. I understand the basic observation, but I found the evidence in this section hard to follow and therefore I didn’t accept the conclusion that evidence for similar TEs in Starships occur because of HTT. The authors themselves admit that “Starships could act as both the source and destination of transposing transposable elements”. Since most cases where different Starships share identical TE are from the same genus, the alternative model of ancestral TE insertion into the Starship and Starship into the host and vertical transmission seems plausible. In the case shown in Fig 6, I struggled to understand how the most parsimonious explanation is for HTT. Since flanking regions for SBS000504 and SBS0000452 are not show, it is not possible to assess the orthology of these insertion locations. I would suggest the authors clearly state the predictions of the alternative evolutionary hypothesis at the beginning of this section, then present the evidence in an unbiased manner that doesn’t treat the conclusion as a given, then assess which alternative hypothesis the evidence best supports.

In terms of broader accessibility to the work, there is very little background provided about the species in question. The introduction focuses almost entirely on TEs and HTT, but provides little context about the biology, phylogeny, and mechanisms of TE controls in *Metarhizium*, which are essential knowledge for interpreting the current results. For example, it is asserted that E6 and NE are from the same species, but no evidence is provided if this is the case. Likewise, Fig 2A suggest that there is paraphyly or taxonomic uncertainty among some *Metarhizium* strains related to *M. anisopliae*.

On a related note, while the introduction focuses on HTT and rightly states that published cases for HTT are rare, they do not review the several known cases of HTT in fungi in the *Saccharomyces* clade, where at least 3 cases have been reported for the Ty1 (PMID: 32084126, 34115140), Ty2 (PMID: 15704235, 23226439), and Ty4/Tsu4 (PMID: 29942366, 39786570) families. In the case of Tsu4 HTT, a very similar pattern of post-HTT expansion and association of the horizontally transferred TE with strain-specific rearrangement breakpoints as seen strain NE has been reported.

Finally, there are a large number of places where call-outs to figures and tables are incorrect in the manuscript, or where

figures are hard to read/interpret. I would encourage a very close edit of this manuscript for proper referencing of figures and tables. I have tried to enumerate these as best I can below, but it is possible I have missed some.

Minor comments:

- p2, line 15: change “serve as sources” to “serve as potential sources”
- p3, line 52: add cases of HTT from *Sacharomyces*
- p3, line 64: add introgression as mechanism for HTT in fungi as implicated in *Sacharomyces*
- p4, line 85: add more background on *Metarhizium* biology, evolution, and mechanisms of TE control
- p4, line 98: delete “618”
- p5, line 109-111: change “chromosomes” to “assembled chromosomes”. (As is, it can read as if several chromosomes lack telomeres in vivo)
- p5, line 123: change “Table 1” to “Table S1”
- p5, line 124: change “Fig 1A,C” to “Fig 1B,D”
- p5, line 124: change “codon usage patterns” to “codon usage patterns (Fig 1C).”
- p5, line 129: provide evidence or citation to back up claim that E6 and NE are from the same species.
- p7, line 164: delete “the presence of one of three individual TE”
- p7, line 175-6: change “these results underscore a significant contribution of transposable elements to the formation of syntenic breaks” to “these results underscore that transposable elements may contribute to the formation of syntenic breaks” (the authors have only shown an association/enrichment of TEs at breakpoints. They have not shown TEs are causal)
- p8, line 193: change “Fig 3A,B” to “Fig 3B,C,D”
- p9, lines 198-200: Conclusion for HT is provided before any evidence is presented. Please reword.
- p9, line 202: change “the NE genome” to “the NE genome (Fig 3C)”
- p9, line 203: “the breakpoints are clearly distinct from other Helitron TEs present in the NE genome”. I am unable to see where breakpoint-associated TEs are in Fig 3C or Fig S4A. Please annotate breakpoint-associated TEs on these figures.
- p9, line 212: change “suggests” to “could be explained by”
- p9, line 213: “we tested for the presence of active Starships in the NE and E6 strains”. Please clarify how this was done. The methods are lacking for how Starships were identified in the newly sequenced genomes.
- p9, line 216: change “Fig S3” to “Fig S5”
- p9, line 218: change “Fig S4A-B” to “Fig S5A-B”
- p9, line 219: change “Fig S4B” to “Fig S5C”
- p9, line 224: change “Fig S4E” to “Fig S5E”
- p9, lines 231-232: Conclusion for Starship mediated transfer is provided before any evidence is presented. Please reword.
- p9, line 235: change “Mutator family” to “Mutator family (Fig 3E)”
- p10, line 244: reword “originally located on Starship s00261 were now dispersed throughout the NE genome”. This statement conflates observation with interpretation.
- p10, line 247: Fisher’s exact test is likely the inappropriate statistical test. Clarify how data fit the assumptions of this test or apply proportion test.
- p10, line 258: change “locate” to “located”
- p12, line 295: change “and we identified” to “and thus we measured”
- p12, line 313: change “Fig 4D” to “Fig 4C”
- p13, line 322: change “local genomic environment” to “local genomic environment of many genes”
- p13, line 338: bold “D)”
- p14, line 349: change “TE families” to “predicted TE families”
- p14, line 374: Fisher’s exact test is likely the inappropriate statistical test. Clarify how data fit the assumptions of this test or apply proportion test.
- p15, line 383: change “light grey” to “dark grey”
- p18, line 455: change “show” to “provide evidence”
- p18, line 499: cite relevant papers on conjugative plasmids
- p19, line 510: change “Horizontal Transposon Transfer (HTT)” to “HTT”
- p19, line 539: rephrase “Starship-derived DNA transposons”. The evidence is not conclusive in my assessment.
- p19, line 541: change “*Metarhizium anisopliae*” to “*M. anisopliae*”
- p19, line 544: cite PMID: 29942366 in the context of HTT leading to rearrangements and reproductive isolation.
- p20, line 549-550: change “most likely now reproductively isolated” to “possibly reproductively isolated”
- p20, line 560: remove period after “fungi:”
- p21, line 587: change “Hongkong” to “Hong Kong”
- p21, line 587: clarify if PacBio data was CLR or HiFi reads
- p21, line 597: change “After” to “Subsequently”
- p23, line 665: explain “BH” acronym.
- p24, line 692 (2X): change “telomers” to “telomeres”
- p25, line 761: explain “LLMs” acronym.
- p28: fix spelling in title of panel D)
- p29, line 805: change “Fig. 2” to “Fig. S2”
- p30, line 812: state what kind of tree is shown and what scale bar represents
- p30, line 812: label rRNA gene cluster
- p32, line 812: state what scale bar represents

Version 1:

Reviewer comments:

Reviewer #1

(Remarks to the Author)

The author revisions have addressed all of my original critiques. I am particularly impressed with the extra analyses they included to test whether Starship TE density differs more than expected from the genomic background using a permutation model.

Reviewer #2

(Remarks to the Author)

Griem-Krey et al. "Transposable elements hitchhike on Starships across fungal genomes" revision 1.

This revised manuscript addresses most of my major concerns from the initial submission, and I appreciate the effort the authors have put into strengthening the major claim that Starships are vectors for HTT in fungi.

The following comments on the current version mainly relate to clarifying new sections of the manuscript and can be considered minor/discretionary.

- p4, line 111: clarify why asexual life cycle would prevent RIP from action on TEs in *Metarhizium*
- p4, line 118: delete "most likely"
- p8, line 179: change "large-scale" to "substantial"
- p9, AFig 2: light grey arrowheads in 2B are very hard to see. Also, the arrowhead for mutator elements looks more grey than brown to me in the pdf, and especially on the printout. I would improve/differentiate the arrowhead colors for the "Mutator" and "none" categories.
- p9, line 215: change "the structural rearrangement" to "structural rearrangements"
- p10, line 236: change "been horizontally" to "been concurrently horizontally"
- p10, line 244: it is stated that *M. anisopliase* NE has 9 starships, but in the introduction it is stated that starships are generally found as single copies (p4, line 88). It would be worth commenting on this discrepancy here and possibly in the discussion since the received wisdom stated in the introduction may need to be updated based on the current study.
- p11, line 280: list families "noted above" since I was not sure which ones were being referred to.
- p11, line 300: clarify what TEs in Starship s00261 are "phylogenetically separate" from.
- p14, line 372: insert a paragraph break before "To test whether..."
- p16, line 425: clarify why there is a switch between using data from EDTA to using data from Earl Grey
- p16, line 451: delete "and observed"
- p18, line 487: italicize *Aspergillus* and *Metarhizium*
- p18, line 496: add "(" before Supplementary
- p18: I would consider combining sections "Starships from different species contain identical TE copies" and "Starships contain TEs that show a high likelihood of past horizontal transfer" and using the latter title for the whole combined section. I was left wondering what the conclusion was to the former section on its own and whether the observation supported vertical or horizontal transfer.
- p19, line 517: delete "Please also see details in Methods"
- p21, line 549: refer to prior analysis on single *M. anisopliase* starship and clarify how the current analysis differs from the prior analysis as part of the set up to this expanded analysis. I initially thought I was reading a duplicate section of text.
- p22, line 599: I would note that the observation that the reverse observation of TEs hopping onto starships is actually essential to the overall model, otherwise there would be no TEs on starships for them to deliver by HT.
- p22, line 608: list which TE families are being referred to in the "four and nine" statement.
- p22, line 611: change "related taxa" to "related fungal taxa"
- p22, line 612: I wonder if it worth speculating that the high rate of HTT in *Mucoromycotina* might imply that there are undiscovered starships or starship-like element in this subphylum.
- p22, line 612-615: italicize taxonomic names
- p24, line 693: change "new" to "incipient"
- p25, line 721: remove superscript from "0."
- p25, line 730: state what library kit was used and how many SMRT cells per sample were generated.
- p27, line 805: change "in this article" to "here"
- p27, line 831: state how assembly statistics were computed.
- p28, lines 845-861: The genome annotation methods are not presented in coherent order and lacking in crucial details about options/parameters. From the text presented, it would be impossible to reproduce how genome annotation was performed. I would encourage the authors to revise this section to allow others to reproduce the findings presented here or make code available.
- p29, lines 883-886. Revise this text to state what is presented rather than what changed from a previous version.
- p34: add scale bar to Sup Fig 1 panel b.

Dear Editors,

Thank you very much for your continued interest in our manuscript “Transposable elements hitchhike on Starships across fungal genomes” and for the opportunity to submit a revised version.

We thank the reviewers for their comments and have addressed each of the points raised. Below, we provide our responses and a summary of the changes to the manuscript. Please note that the reviewers' comments are in blue and italics, while our responses are in black. We have attached both a clean manuscript file and a manuscript file with tracked changes. The line numbers in our reply refer to our new manuscript version with fully-displayed tracked changes.

Please note: The Starship accessions have changed in the updated STARBASE repository. To facilitate future comparisons, we have updated all accessions for the Starships used in this study and revised all mentions of them in the text, figures, and tables. A mapping of the old versus new accessions is provided in Table S4.

We hope that you find our revised version suitable for publication in Nature Communications and look forward to hearing from you.

With kind regards,

Michael Habig on behalf of all authors

Reviewer #1 (Remarks to the Author):

In this manuscript, Griem-Krey et al. investigate the phenomenon of horizontal transposon transfer (HTT) in fungal microbes. They find that giant Starship transposons are likely a common vector of HTT, and that the introduction of transposons into new genomes has important impacts on genome evolution and the evolution of fungal pathogenicity. Crucially, the authors enhance their genomic analyses with infection assays and secretome analyses, and identify interesting associations between observed decreases in pathogenicity and structural rearrangements induced by TE expansion. The authors then take the additional step to generalize their findings by testing for TE-enrichment in a larger database of Starship sequences. The manuscript contributes a timely and impactful story to the literature but there are a couple of opportunities for improvement. Below I outline several suggestions that I believe are necessary to strengthen and support the authors' claims of HTT and TE enrichment on Starships.

Please note: Starship accessions have changed in the updated STARBASE repository. To facilitate future comparisons, we have updated all accessions for the Starships used in this study and revised all mentions of them in the text, figures, and tables. Because the new accessions in STARBASE categorize previously separate Starships as identical, we removed duplicate Starships from the analysis. This reduced the number of analyzed published Starships from 618 to 522. A mapping of the old versus new accessions is provided in Table S4, which also indicates which Starships are now considered duplicates and were therefore excluded from the analysis.

The in text citations for the supplementary figures do not match the figure legends (e.g. in text, Fig S4 actually appears to refer to Fig S5...)

We have checked and corrected all in text citations.

Line 50: more clear if you change to "to escape a specific host genome's..."

Done

L98: missing a space between "known 618"

Done

L224, Fig S5E: were only protein sequences aligned to make this synteny alignment? I think it would be more insightful if the alignments were based on nucleotide sequences, that way you could account more precisely for sequences that are shared/divergent between these starships

We updated the phylogeny from protein-based alignments to nucleotide alignments of predicted genes (Supplementary Fig. 5d). The overall topology is similar: within NE, four Captain genes (g6244, g7476, g8301, and g7614) cluster tightly, whereas the Starship s00261 Captain (g6294) lacks close homologs in E6 and other *M. anisopliae* strains; its closest phylogenetic relative is from *M. humberi*. Additional details on the phylogenetic relationships of the s00261 Captain, Starship genes, and transposable elements are provided in Supplementary Fig. 6a-b (see also comments below; lines 1170ff).

We also replaced the gene-based synteny alignment in Supplementary Fig. 5e with a nucleotide-based synteny analysis spanning the entire Starship and its flanking regions (not restricted to coding sequences). The results are very similar, and the conclusion remains unchanged: the Starships share synteny, but their insertion sites do not. We have updated Supplementary Fig. 5e accordingly (lines 1149ff).

L232: TE acronym has already been defined

Done

L231-240: I'm confused. How many TE copies are on Starship s00261? 73? And does this include the 9 copies of the TE families mentioned earlier? I think I understand, but this is a really crucial result so please clarify as best you can

NE contains nine Starships, including s00261; s00261 harbors 73 annotated TEs. Of these 73, three belong to the expanded Helitron family, four to the hAT-Restless family, and two to the Mutator family. These families were also present at TE-associated synteny breakpoints in the NE genome. We clarified this in lines 296ff and have included the appropriate figure caption.

*L248: why would an increased density of TEs on the Starship indicate that it is a source? I guess this assumes that the TE landscape in whatever the donor genome is has a relatively high density of these TE insertions such that they accumulated in the Starship region. Could be good to be more transparent with the logic here. Also, since genomes of *M. pinghaense* and *M. robertsii* are available, why not actually test whether the density of the TEs on the Starship matches the genomic background TE density of these other species? This would strengthen your argument.*

Based on this comment and those from Reviewer 2, we re-analyzed our data in two areas: 1) the potential association between Starships and transposable elements (TEs), and (2) the age of TEs on Starships, the fraction of expanded TEs on Starships, and whether TEs on Starships may be a source of expanding TEs.

For aspect 1), we agree that a higher TE content within Starships does not in itself show that Starships are the source of TEs; it indicates an association between Starships and TEs. We therefore asked whether Starships are comparable in TE content to their host genomes using two complementary approaches: (i) a window-based comparison under a “random TE placement” assumption, and (ii) an interval-permutation test under a “random Starship placement” assumption that preserves each genome’s empirical TE landscape. The first test assesses whether TE content within a Starship differs from genome-wide TE content; the second evaluates whether TE content within Starships is more extreme than that of randomly placed, Starship-sized intervals in the same genome.

Across 352 Starships (after multiple-testing correction), 26 showed significant deviations from genomic background in the window-based test (23 enriched, 3 depleted for TE density). In the permutation framework, 6 Starships had significantly higher TE density than expected under random placement. Together, these results indicate that a subset of Starships differs markedly from the surrounding genome - some ranking among the most TE-rich regions (see further description of approach in Methods lines 1020ff and in Results in lines 450ff). While this demonstrates a robust association between Starships and TE accumulation, it does not by itself establish whether Starships act as a source or a sink for TEs

To address this question, we evaluated (i) the relative ages of TE copies on Starships versus elsewhere in the genome, (ii) the proportion of TE families showing recent expansion on Starships versus outside Starships, and (iii) TE copy numbers. For several Starships we observed: (i) some TE copies on Starships are older than those in the genomic background; (ii) a higher proportion of expanded TE families on Starships; and (iii) larger copy numbers among those expanded families on Starships. Together with our exemplar analyses of horizontal TE transfer in the genera *Aspergillus* and *Metarhizium* - which show that a high fraction of TEs on Starships bear signatures of horizontal transfer - these findings collectively support the more likely, parsimonious scenario that TEs were horizontally acquired via Starships and subsequently expanded. We have updated these results (lines 543ff, lines 591ff) and revised the Discussion (line 637ff).

L209-231: I don't see evidence supporting the horizontal transfer of the Starship itself, beyond the analysis looking at signatures of horizontal transfer in the TE sequences. Yet the authors refer to "horizontally acquired Starships" (L231). If you want to call the Starship horizontally transferred, it would strengthen your argument if you looked for evidence of its presence in other fungi, especially the putative donors M. pinghaense or M. robertsii, beyond the individual TE sequences located on the Starship (like, what about the captain or other genes? Do they have signatures of HGT too?). Throughout the description of these results, it may also be helpful to mention the taxonomic scope of the databases that were searched, for example, when looking for the closest relative to the Starship-borne TEs. Did this database contain only Metarhizium species or a more taxonomically diverse group? How many isolates per species?

For Starship s00261, we analyzed the phylogenetic relationships of all 124 genes annotated on this element - placing particular emphasis on the Captain gene - across species of the genus *Metarhizium* (Supplementary Figure 6). We observe presence/absence polymorphism of the Captain gene among closely related *M. anisopliae* strains. The Captain's closest ortholog occurs in *M. humberi* (strain ESALQ1638) and is absent from the closely related *M. anisopliae* strains examined. In addition, the Captain alignment rejects the species phylogeny inferred from 347 single-copy BUSCO genes, further supporting horizontal transfer. Among the 123 cargo genes of this Starship, 108 either reject the BUSCO-based species phylogeny across these *Metarhizium* species or are unique to Starship s00261 (i.e., lack detectable orthologs in any surveyed *Metarhizium* genome). We therefore conclude that Starship s00261 was horizontally acquired by *M. anisopliae* NE (see lines 267ff). To clarify taxonomic scope (10 different *Metarhizium* species with a total of 20 isolates), we have included the list of species and corresponding genome accessions used in this and all other analyses (Supplementary Tables 4 and 8) and also mentioned this in the text (line 274).

L374-376: one issue with the null hypothesis that TEs are randomly distributed across the genome is that we know that this is almost certainly never the case because TEs tend to accumulate in TE-rich compartments in many fungal genomes. The question of whether TEs are enriched in Starships or whether Starships tend to be in TE-rich compartments is a bit of a chicken and egg question. The key comparison that needs to be done in my opinion is to test whether the pattern of TE enrichment observed in Starships differs from other TE rich compartments in the genome. For example, can you group TEs into genomic regions (e.g. group all TEs within 10kb of each other into the same region), and then test for enrichment of particular TEs across all these regions? Are Starships the only such region that pop out of the analysis, or are there others as well? Determining whether Starships are somehow "special" would go a long way towards strengthening your argument that they are the source of TEs.

To account for the non-random distribution of transposable elements (TEs) within genomes, we implemented a permutation test that uses each genome's actual TE distribution. Specifically, we

permuted Starship-sized windows along the genome and compared the TE content of each Starship to this null distribution. For seven Starships, we find significant TE enrichment - i.e., their TE content is more extreme than elsewhere in the genome - indicating that these Starships are exceptional or “special”. Together with our additional new results showing that (i) many TEs on Starships exhibit signatures of horizontal transfer, including identical elements in phylogenetically distinct species; (ii) some copies of expanding TEs on Starships are older than those elsewhere in the genome, suggesting Starships could be the source of expanding TEs; (iii) a higher proportion of TEs on Starships are expanded; and (iv) these expanded TEs have larger copy numbers, the most parsimonious explanation is that TEs were horizontally acquired on Starships and subsequently expanded. We have updated these results (lines 543ff, lines 591ff) and revised the Discussion (line 637ff).

L408-411, L415, L424: Comparing A. oryzae and A. flavus as different “species” is not the most convincing, as A. oryzae is thought by some to not be a separate species at all but a domesticated A. flavus lineage (the two “species” share 99.5% genome-wide nucleotide similarity). I honestly don’t think its worth highlighting, because it detracts from your argument, and I don’t think you can confidently claim that similar Starships found in these two “species” were exchanged via HTT. Is there a better example? Whatever example you do pick, please make sure the species boundaries are well supported in the literature.

We updated species assignments for *Aspergillus* spp. based on the revised phylogeny reported by Steenwyk and colleagues. In addition, we corrected the species identifications for 29 Starships that were associated with misidentified species in the NCBI repository, following Steenwyk et al.¹. Because *A. flavus* and *A. oryzae* exhibit very low divergence and are frequently misidentified - consistent with the reviewer’s comments - we now follow Steenwyk et al. in referring to these collectively as “*A. flavus* or *A. oryzae*”. In addition, we generated a species phylogeny for all *Aspergillus* isolates using 411 single-copy BUSCO orthologs and confirmed that *A. flavus* and *A. oryzae* are not phylogenetically separated (Supplementary Fig. 10). Accordingly, we no longer treat TE transfers between these two taxa as interspecific events and have removed this example. In response to Reviewer 2, we analysed horizontal transfer of Starship TEs in greater detail - exemplified by Starships within the genera *Metarhizium* and *Aspergillus* - and found evidence of horizontal transfer for many of them, sometimes for all TEs (>1 kb), on a Starship. We have replaced parts of the previous analysis with this more in-depth treatment (lines 543ff; Fig. 6; Supplementary Fig. 11).

Can you please include estimates of how often you observe at least one TE with a perfect copy on a Starship from a different genus? You mention some on L478-479, these would be most interesting to see. This would rise above some of the issues of identifying HTT among closely related species.

We identified four TEs with identical copies in Starships from species of different genera, plus nine additional TEs with >99% identity and coverage. We have included these counts in the text (line 663).

L680: the URL does not work when I copy and paste, please provide full URL

Done

Reviewer #2 (Remarks to the Author):

Griem-Krey et al. "Transposable elements hitchhike on Starships across fungal genomes"

*This manuscript reports genome assemblies and annotations for 2 strains of the insect-pathogenic fungus *Metarhizium anisopliae*, and a comparative analysis of transposable element (TE) content in these strains and related species. The central claim of the paper is that a class of newly identified large TEs called Starships that undergo horizontal transfer (HT) can carry other smaller TEs that can infect new genomes and impact genome organization.*

My major concern with the manuscript is that the evidence in support of the claim that Starships are vectors for horizontal transposon transfer (HTT) is reasonably good, but not as strong as the authors assert in their manuscript. There are three well-established criteria enumerated by Wallau et al (PMID: 22798449) for showing HTT, which the authors have not fully shown to prove HTT. Additionally, in many places the authors state the conclusion for Starship-driven HTT as proven, before providing evidence in support of this conclusion or considering alternative hypotheses. For this manuscript to not mislead a general readership about the strength of the evidence for the stated claims, the evidence for Starship-driven HTT needs to be framed in terms of Wallau et al's criteria and carefully evaluated with respect to alternative hypotheses prior to making the major conclusion, which I believe would lead to a more accurate and balanced statement of the support for the authors claims.

One specific area of concern I have related to the proof that Starships are donors of HT TEs is the statistical analysis of TEs that share high identity inside and outside Starships. If I understand the argument correctly, if Starships are the donors for TEs that undergo post-HTT expansion there should be a higher density of TEs in Starships relative to the recipient genome. The authors use Fisher's exact test to provide evidence for this prediction. I have two issues with this analysis. First, Fisher's exact test should be used for numbers from two categories in a 2x2 contingency test for association. The authors do not provide enough information to understand how TE density inside and outside starships fits this the requirements for this test. In my opinion, this test is inappropriate, and the appropriate test should be a proportion test.

We performed several additional analyses to test three aspects:

1. Association of Starships with TEs (this section);
2. Whether TEs on Starships serve as sources/donors of expanding TEs (next section).
3. Horizontal transfer of TEs on Starships (following section of this response letter)

1. Association of Starships with TEs: To test whether Starships differ from the rest of the genome in TE content, we annotated TEs in all Starships and their available genome accessions (363 Starships in 193 genomes). Note that Fisher's exact and two-proportion tests assume independent trials; applied at base-pair resolution, this assumption is violated because whether a base is part of a TE is strongly correlated with the state of neighboring bases. Therefore, instead of these tests we used two complementary approaches to test whether Starships differ in TE content from the rest of the genome:

- Window-based comparison under random TE placement. We split each genome into non-overlapping 2-kb windows, computed TE coverage per window, randomly selected ~1/3 of Starship windows and the same number of non-Starship windows (to reduce spatial autocorrelation and balance sample sizes), and compared the two distributions using a two-sided

Wilcoxon rank-sum test, and p-values were FDR-adjusted across Starships. This interpretation assumes no strong clustering of TEs outside Starships.

- Interval permutation using the empirical TE landscape. For each Starship, we held its length fixed and generated a null distribution by randomly relocating an interval of identical length to non-Starship regions of the same genome (100,000 permutations), recomputing TE overlap each time. Two-sided p-values were derived from the empirical null. This preserves genome architecture and the observed TE landscape outside Starships while testing whether the observed TE content inside a Starship deviates from expectation under random placement.

Using both tests, we find that a subset of Starships differs significantly from the genomic background in TE content: 23 are enriched and 3 are depleted relative to the rest of the genome, with six Starships showing significantly higher TE density than expected under random Starship placement. These results demonstrate an association between Starships and TE presence; however, on their own they do not establish causality (i.e., whether Starships act as sources or sink for expanding TEs).

Please note: Starship accessions have changed in the updated STARBASE repository. To facilitate future comparisons, we have updated all accessions for the Starships used in this study and revised all mentions of them in the text, figures, and tables. Because the new accessions in STARBASE categorize previously separate Starships as identical, we removed duplicate Starships from the analysis. This reduced the number of analyzed published Starships from 618 to 522. A mapping of the old versus new accessions is provided in Table S4, which also indicates which Starships are now considered duplicates and were therefore excluded from the analysis.

Moreover, regardless of the appropriateness of the statistical test used, the result of this hypotheses test does not definitively support Starships as donors, since the alternative hypothesis of Starships being less constrained at the nucleotide level than the host genome (and hence more likely to tolerate new TE insertions) is an equally plausible interpretation of the data.

2. Whether TEs on Starships serve as sources/donors of expanding TEs. To address this question, we evaluated (i) the relative ages of TE copies on Starships versus elsewhere in the genome, (ii) the proportion of TE families showing recent expansion on Starships versus outside Starships, and (iii) TE copy numbers. For several Starships we observed: (i) some TE copies on Starships are older than those in the genomic background; (ii) a higher proportion of expanded TE families on Starships; and (iii) larger copy numbers among those expanded families on Starships. Together with our exemplar analyses of horizontal TE transfer in the genera *Aspergillus* and *Metarhizium* (see next section of this response letter) - which show that a high fraction of TEs on Starships bear signatures of horizontal transfer - these findings collectively support the more likely, parsimonious scenario that TEs were horizontally acquired via Starships and subsequently expanded. We have updated these results (lines 543ff, lines 591ff, Supplementary Fig. 7 and 12) and revised the Discussion (line 637ff).

Another specific area of concern relates to the last section of the results “Starships from different species contain identical TE copies indicating HTT between species”. I understand the basic observation, but I found the evidence in this section hard to follow and therefore I didn’t accept the conclusion that evidence for similar TEs in Starships occur because of HTT. The authors themselves admit that “Starships could act as both the source and destination of transposing transposable elements”. Since most cases where different Starships share identical TE are from the same genus, the alternative model of ancestral TE insertion into the Starship and Starship into the host and vertical transmission seems plausible. In the case shown in Fig 6, I struggled to understand how the most parsimonious explanation is for HTT. Since flanking regions for SBS000504 and SBS0000452 are not

show, it is not possible to assess the orthology of these insertion locations. I would suggest the authors clearly state the predictions of the alternative evolutionary hypothesis at the beginning of this section, then present the evidence in an unbiased manner that doesn't treat the conclusion as a given, then assess which alternative hypothesis the evidence best supports.

3. Horizontal transfer of TEs on Starships. We performed additional analyses to test whether TE lineages on Starships reflect horizontal transfer versus vertical transmission. Following Wallau *et al.* ², we evaluated three criteria for HTT: (i) host - TE phylogenetic incongruence, (ii) patchy TE distribution across species, and (iii) high interspecific TE sequence similarity. We conducted this analysis for all Starships in the genera *Metarhizium* and *Aspergillus*.

For each genus, we inferred single-copy BUSCO - based species phylogenies (347 and 411 single-copy orthologs for *Metarhizium* spp. and *Aspergillus* spp., respectively). Using the corresponding single-copy BUSCO alignments, we estimated the rate of synonymous substitutions per synonymous site (dS) for all pairwise genome comparisons within each genus. For every TE >1 kb located on a Starship, we blasted the TE sequence in all isolates within the genus (including outgroups) using relaxed thresholds ($\geq 80\%$ identity and $\geq 80\%$ coverage) to determine presence/absence. To test phylogenetic incongruence, we compared TE alignments that met these thresholds against the BUSCO species trees using the AU test³; rejection of the species tree indicated distinct TE versus host histories which would be consistent with HTT.

To evaluate sequence similarity, we restricted comparisons to species (or strain) pairs with genome-wide dS ≥ 0.02 (to ensure stable distance estimates and detection of HTT). Within those pairs, we considered HTT likely when the entire TE was identical between species or, if not identical, when the largest open reading frame (ORF) within the TE had dS(TE) $\leq 10\%$ of the corresponding species-pair dS(species).

Applying these criteria, in *Aspergillus* we found that 15% (25/171) of Starships contain at least one TE with a high likelihood of horizontal transfer; in *Metarhizium*, the fraction is 33% (13/39). At the level of individual Starships, several harbor a high proportion of such TEs. For example, all nine TEs (>1 kb) in SSA003776 (Old accession: SBS000559) meet these criteria, with identical or very closely related TE sequences present in this Starship from *A. luchuensis* and in the distantly related *A. niger*.

Together, these lines of evidence indicate that multiple Starship-associated TEs likely have undergone horizontal transfer, and in some cases this involves a substantial share of all TEs (>1 kb) on a given Starship. We have updated our manuscript accordingly and included these results in line 543ff, Fig. 6 and Supplementary Fig. 11.

We replaced the original Figure 6 because our new analysis is more informative, and the previously interpreted case of interspecific horizontal TE transfer between *A. oryzae* and *A. flavus* is ambiguous with respect to species boundaries. In line with Reviewer 1's comments, the phylogeny of Steenwyk *et al.* ¹, and our own phylogeny based on 411 single-copy BUSCO orthologs for *Aspergillus* spp., we find no robust separation between *A. flavus* and *A. oryzae*. Accordingly, we no longer interpret this Starship transfer - or any transfers between *A. flavus* and *A. oryzae* - as interspecific. All corresponding figures and tables have been updated to reflect that change.

In terms of broader accessibility to the work, there is very little background provided about the species in question. The introduction focuses almost entirely on TEs and HTT, but provides little context about the biology, phylogeny, and mechanisms of TE controls in Metarhizium, which are essential knowledge

for interpreting the current results. For example, it is asserted that E6 and NE are from the same species, but no evidence is provided if this is the case. Likewise, Fig 2A suggest that there is paraphyly or taxonomic uncertainty among some Metarhizium strains related to M. anisopliae.

Indeed, there is some uncertainty within *Metarhizium* spp., and the basis for the phylogenetic species assignments of some genome assemblies at NCBI is unclear. Accordingly, we re-ran a phylogenetic analysis of all *Metarhizium* isolates used in this study, updating it to use DNA sequences from 347 single-copy BUSCO orthologs (see Supplementary Fig. 2a,b) in place of the earlier protein-based phylogeny. This analysis shows that the *M. anisopliae* NE and E6 isolates are monophyletic and form a clade with other *M. anisopliae* isolates, with high branch support. It also enabled us to identify isolates that, presumably classified on the basis of a few genetic markers, were misidentified in the NCBI repository: *M. pinghaense* JEF157 (deposited as *M. anisopliae* JEF157) and *M. brunneum* JEF290 (deposited as *M. anisopliae* JEF290), similar to the detection of misidentified *Aspergillus* isolates using single-copy BUSCO analyses reported by Steenwyk et al.¹ We therefore conclude that E6 and NE are conspecific based on the gene-based phylogeny, in contrast to the pronounced differences in macrosynteny between these two isolates. We have updated the corresponding figure caption and species identifications for the misidentified isolates throughout the text. We have included information on *Metarhizium* spp. biology with a focus on what is known about its TEs and genomic defenses - which is, however, very limited (line 104ff).

On a related note, while the introduction focuses on HTT and rightly states that published cases for HTT are rare, they do not review the several known cases of HTT in fungi in the Saccharomyces clade, where at least 3 cases have been reported for the Ty1 (PMID: 32084126, 34115140), Ty2 (PMID: 15704235, 23226439), and Ty4/Tsu4 (PMID: 29942366, 39786570) families. In the case of Tsu4 HTT, a very similar pattern of post-HTT expansion and association of the horizontally transferred TE with strain-specific rearrangement breakpoints as seen strain NE has been reported.

We thank the reviewer for these comments. We were not aware of these reports, and the case of a horizontally transferred TE associated with strain-specific rearrangements and (partial) reproductive isolation in *Saccharomyces paradoxus* shows intriguing parallels to our findings. We have therefore incorporated this example in both the Introduction (line 66ff) and the Discussion (lines 741ff).

Finally, there are a large number of places where call-outs to figures and tables are incorrect in the manuscript, or where figures are hard to read/interpret. I would encourage a very close edit of this manuscript for proper referencing of figures and tables. I have tried to enumerate these as best I can below, but it is possible I have missed some.

Minor comments:

- p2, line 15: change “serve as sources” to “serve as potential sources”

Done

- p3, line 52: add cases of HTT from *Sacharomyces*

Done

- p3, line 64: add introgression as mechanism for HTT in fungi as implicated in *Sacharomyces*

Done

- p4, line 85: add more background on *Metarhizium* biology, evolution, and mechanisms of TE control

Done (line 104ff) – however knowledge on the mechanism of TE control in *Metarhizium* spp. is very limited.

- p4, line 98: delete “618”

Done

- p5, line 109-111: change “chromosomes” to “assembled chromosomes”. (As is, it can read as if several chromosomes lack telomeres in vivo)

Done

- p5, line 123: change “Table 1” to “Table S1”

Done

- p5, line 124: change “Fig 1A,C” to “Fig 1B,D”

Done

- p5, line 124: change “codon usage patterns” to “codon usage patterns (Fig 1C).”

Done

- p5, line 129: provide evidence or citation to back up claim that E6 and NE are from the same species.

We have included a phylogeny based on DNA sequences from 347 single-copy BUSCO orthologues, which shows that NE and E6 are monophyletic (Supplementary Fig. 2a,b) and rephrased line 178ff accordingly.

- p7, line 164: delete “the presence of one of three individual TE”

Done

- p7, line 175-6: change “these results underscore a significant contribution of transposable elements to the formation of synteny breaks” to “these results underscore that transposable elements may contribute to the formation of synteny breaks” (the authors have only shown an association/enrichment of TEs at breakpoints. They have not shown TEs are causal)

Done

- p8, line 193: change “Fig 3A,B” to “Fig 3B,C,D”

Done

- p9, lines 198-200: Conclusion for HT is provided before any evidence is presented. Please reword.

Done

- p9, line 202: change “the NE genome” to “the NE genome (Fig 3C)”

Done

- p9, line 203: *“the breakpoints are clearly distinct from other Helitron TEs present in the NE genome”. I am unable to see where breakpoint-associated TEs are in Fig 3C or Fig S4A. Please annotate breakpoint-associated TEs on these figures.*

Fig. 3c,d and Supplementary Fig. 4a have been revised: expanded TEs in the NE genome now share the same background colour (as in Fig. 3b), whereas non-expanded elements do not, though they remain in bold.

- p9, line 212: change “suggests” to “could be explained by”

Done

- p9, line 213: *“we tested for the presence of active Starships in the NE and E6 strains”. Please clarify how this was done. The methods are lacking for how Starships were identified in the newly sequenced genomes.*

This is briefly described in the Methods section as we strictly followed the procedure already explained in detail in the reference (line 943). Because a Starship can be recognized only when synteny comparisons between two genome assemblies show that it has moved, we have clarified this prerequisite in the text (lines 247ff.)

- p9, line 216: change “Fig S3” to “Fig S5”

Done. Please note that the Supplementary Figures have been renamed according to the requirements of Nature Communications (e.g., changing 'Fig S3' to 'Supplementary Figure 3').

- p9, line 218: change "Fig S4A-B" to "Fig S5A-B"

Done

- p9, line 219: change "Fig S4B" to "Fig S5C"

Done

- p9, line 224: change "Fig S4E" to "Fig S5E"

Done

- p9, lines 231-232: Conclusion for Starship mediated transfer is provided before any evidence is presented. Please reword.

Done

- p9, line 235: change "Mutator family" to "Mutator family (Fig 3E)"

Done

- p10, line 244: reword "originally located on Starship s00261 were now dispersed throughout the NE genome". This statement conflates observation with interpretation.

Done

- p10, line 247. Fisher's exact test is likely the inappropriate statistical test. Clarify how data fit the assumptions of this test or apply proportion test.

Done. We agree that Fisher's exact test is inappropriate and have applied two different tests - one assuming random TE locations and one assuming random Starship locations. Please see the updated manuscript text (lines 446ff) and our earlier comments in this response letter regarding the suitability of these tests.

- p10, line 258: change "locate" to "located"

Done

- p12, line 295: change "and we identified" to "and thus we measured"

Done

- p12, line 313: change "Fig 4D" to "Fig 4C"

Done

- p13, line 322: change "local genomic environment" to "local genomic environment of many genes"

Done

- p13, line 338: bold "D)"

Done

- p14, line 349: change "TE families" to "predicted TE families"

Done

- p14, line 374: Fisher's exact test is likely the inappropriate statistical test. Clarify how data fit the assumptions of this test or apply proportion test.

Please see our earlier comments on the inappropriateness of Fisher's exact test and our approach to statistically test for the association between Starships and TEs. We added analyses of i) copy divergence from the consensus, ii) the fraction of expanded TEs, and iii) the number of TE copies, each evaluated by whether the copy is located inside or outside a starship (please see earlier comments). Accordingly, we have removed the Fisher's exact test results.

- p15, line 383: change "light grey" to "dark grey"

Done

- p18, line 455: change "show" to "provide evidence"

Done

- p18, line 499: cite relevant papers on conjugative plasmids

Done - on the conceptual similarity between Starships and conjugative plasmids.

- p19, line 510: change "Horizontal Transposon Transfer (HTT)" to "HTT"

Done

- p19, line 539: rephrase "Starship-derived DNA transposons". The evidence is not conclusive in my assessment.

Done. We have rephrased this to now read. "... of these likely Starship-derived DNA transposons."

- p19, line 541: change "Metarhizium anisopliae" to "M. anisopliae"

Done

- p19, line 544: cite PMID: 29942366 in the context of HTT leading to rearrangements and reproductive isolation.

We have now included the following statement in the discussion (line 741ff): Such an HTT, followed by TE proliferation that induced genome rearrangements and in turn contributed to (partial) reproductive isolation, has already been reported in *S. paradoxus*⁴

- p20, line 549-550: change "most likely now reproductively isolated" to "possibly reproductively isolated"

Done

- p20, line 560: remove period after “fungi:”

Done

- p21, line 587: change “Hongkong” to “Hong Kong”

Done

- p21, line 587: clarify if PacBio data was CLR or Hifi reads

Done. Hifi reads

- p21, line 597: change “After” to “Subsequently”

Done

- p23, line 665: explain “BH” acronym.

Done. Benjamini–Hochberg

- p24, line 692 (2X): change “telomers” to “telomeres”

Done

- p25, line 761: explain “LLMs” acronym.

Done

- p28: fix spelling in title of panel D)

Done

- p29, line 805: change “Fig. 2” to “Fig. S2”

DONE

- p30, line 812: state what kind of tree is shown and what scale bar represents

Done

- p30, line 812: label rRNA gene cluster

Using Barrnap (v0.9), we updated the rRNA gene cluster annotation, which places the cluster further downstream of the region with high SNP density. The previous annotation - based solely on the position of a (as we now know - partial) 5S rRNA gene - was therefore incorrect. Because this revised location does not affect the result shown in the figure - that variation in SNP/INDEL density does not correspond to the mosaic of different syntenic E6 chromosomes across the NE chromosomes - we have removed the reference to the rRNA cluster location from the figure legend.

- p32, line 812: state what scale bar represents

Done

Dear Editors,

Thank you for the acceptance, in principle, of our manuscript, “Transposable elements hitchhike on Starships across fungal genomes”, for publication in *Nature Communications*.

We appreciate the reviewers’ constructive feedback and have addressed each of the points raised. Below, we provide a point-by-point response to the comments and a summary of the resulting changes. For clarity, the reviewers’ comments are shown in blue italics, while our responses are in black. We have attached both a clean version of the manuscript and a version with tracked changes. Please note that the line numbers in our responses refer to the version with tracked changes. We hope the revised manuscript is now suitable for publication and look forward to hearing from you.

Sincerely,

Michael Habig, on behalf of all authors

REVIEWERS' COMMENTS

Reviewer #1 (Remarks to the Author):

The author revisions have addressed all of my original critiques. I am particularly impressed with the extra analyses they included to test whether Starship TE density differs more than expected from the genomic background using a permutation model.

Reviewer #2 (Remarks to the Author):

Griem-Krey et al. "Transposable elements hitchhike on Starships across fungal genomes" revision 1.

This revised manuscript addresses most of my major concerns from the initial submission, and I appreciate the effort the authors have put into strengthening the major claim that Starships are vectors for HTT in fungi.

The following comments on the current version mainly relate to clarifying new sections of the manuscript and can be considered minor/discretionary.

- p4, line 111: clarify why asexual life cycle would prevent RIP from action on TEs in Metarhizium

We have included now a half sentence, That RIP is considered to be only happening during the sexual stages of the life cycle and therefore should not be affecting TEs in these mostly asexual fungi. (now line 93)

- p4, line 118: delete "most likely"

Done (now line 101)

- p8, line 179: change "large-scale" to "substantial"

Done (now line 164)

- p9, AFig 2: light grey arrowheads in 2B are very hard to see. Also, the arrowhead for mutator elements looks more grey than brown to me in the pdf, and especially on the printout. I would improve/differentiate the arrowhead colors for the "Mutator" and "none" categories.

We replaced the 'none' category symbols with open gray triangles to better differentiate them from the 'Mutator' category and updated the caption. (Fig.2, line 1255)

- p9, line 215: change "the structural rearrangement" to "structural rearrangements"

Done. (line 175)

- p10, line 236: change “been horizontally” to “been concurrently horizontally”

Done. (line 196)

- p10, line 244: it is stated that *M. anisopliase* NE has 9 starships, but in the introduction it is stated that starships are generally found as single copies (p4, line 88). It would be worth commenting on this discrepancy here and possibly in the discussion since the received wisdom stated in the introduction may need to be updated based on the current study.

The presence of multiple Starship copies within a single genome has been previously documented in several fungi, for instance, in *Macrophomina phaseolina* (Gluck-Thaler et al., 2022)¹. However, as this remains the exception rather than the rule, we characterized them as typically single-copy in our introduction. Consequently, the identification of multiple copies in the NE strain is not a novel observation and does not warrant further elaboration, as such a discussion would distract from our primary focus: the role of Starships as vectors for TEs.

- p11, line 280: list families “noted above” since I was not sure which ones were being referred to.

Done – specified as Helitron, hAt-Restless and Mutator. (line 241).

- p11, line 300: clarify what TEs in Starship s00261 are “phylogenetically separate” from.

We have specified that these TEs are phylogenetically different from other members of their TE superfamily in other *Metarhizium* spp. isolates. (line 260 ff)

- p14, line 372: insert a paragraph break before “To test whether...”

Done. (line 318)

- p16, line 425: clarify why there is a switch between using data from EDTA to using data from Earl Grey

During the revision, we transitioned from EDTA to Earl Grey to facilitate the annotation of 193 complete fungal genomes (including both Starship elements and the genomic background). This shift was necessitated by the scale of the task; Earl Grey is more computationally efficient and provided greater stability than EDTA within our revision timeframe. To ensure the validity and consistency of our results, we only compared annotations generated by the same software. For example, when comparing the TE content within a Starship to the content outside of it (Fig. 5a, b), we exclusively used

Earl Grey annotations. We have updated the Materials and Methods section (line 744 and lines 756 ff) to specify which software was utilized for each analysis.

- p16, line 451: delete “and observed”

Done. (line 379)

- p18, line 487: italicize *Aspergillus* and *Metarhizium*

Done. (line 395)

- p18, line 496: add “(“ before *Supplementary*

Done. (line 404)

- p18: I would consider combining sections “Starships from different species contain identical TE copies” and “Starships contain TEs that show a high likelihood of past horizontal transfer” and using the latter title for the whole combined section. I was left wondering what the conclusion was to the former section on its own and whether the observation supported vertical or horizontal transfer.

We believe these two paragraphs are appropriate, as they accurately reflect our process for identifying the horizontal transfer of TEs located within Starship elements.

- p19, line 517: delete “Please also see details in Methods”

Done.

- p21, line 549: refer to prior analysis on single *M. anisopliae* starship and clarify how the current analysis differs from the prior analysis as part of the set up to this expanded analysis. I initially thought I was reading a duplicate section of text.

We used the analysis of Starship s00261 (from *M. anisopliae* NE) as a template for investigating transposable elements (TEs) within 25 Starships characterized by large TE expansions. (line 445 ff)

- p22, line 599: I would note that the observation that the reverse observation of TEs hopping onto starships is actually essential to the overall model, otherwise there would be no TEs on starships for them to deliver by HT.

We state that the transposition of a TE onto a Starship is a prerequisite for its future horizontal transfer. (line 461)

- p22, line 608: list which TE families are being referred to in the “four and nine” statement.

We have stated that the four and nine TE Families belong to four TE superfamilies (TC1 Mariner, RLG (Gypsy), Bel Pao LTR and Mutator superfamilies). (line 504)

- p22, line 611: change “related taxa” to “related fungal taxa”

Done. (line508)

- p22, line 612: *I wonder if it worth speculating that the high rate of HTT in Mucoromycotina might imply that there are undiscovered starships or starship-like element in this subphylum.*

We prefer to not speculate about this in our text – but we are aware that people are investigating the distribution of starship-like elements.

- p22, line 612-615: *italicize taxonomic names*

Done. (lines 509 ff)

- p24, line 693: change “new” to “incipient”

Done. (line 587)

- p25, line 721: *remove superscript from “0.”*

Done. (line 615)

- p25, line 730: *state what library kit was used and how many SMRT cells per saple were generated.*

The SMRTbell® Prep Kit 3.0 (PacBio, PN 102-182-700) was utilized for library preparation. The specific number of SMRT cells used was not provided by the service provider (BGI), as their deliverables are based on total data output (which is already specified in the manuscript) rather than cell count. This clarification has been added to the manuscript. (line 623)

- p27, line 805: change “in this article” to “here”

Done. (line 699)

- p27, line 831: *state how assembly statistics were computed.*

The assembly statistics, calculated using Tapestry, are provided in Supplementary Table 1, which also includes the total number of genes and transcripts. (line 725 ff)

- p28, lines 845-861: *The genome annotation methods are not presented in coherent order and lacking*

in crucial details about options/parameters. From the text presented, it would be impossible to reproduce how genome annotation was performed. I would encourage the authors to revise this section to allow others to reproduce the findings presented here or make code available.

The code for the bioinformatics analysis has been made available in a GitHub repository. Please refer to the Code Availability Statement in line 871.

- p29, lines 883-886. Revise this text to state what is presented rather than what changed from a previous version.

To ensure full transparency regarding our dataset, we believe it is essential to describe the process used to obtain the final number of Starships from STARBASE. We therefore prefer to include a detailed account of the initial data retrieval on March 10, 2025, and the subsequent updates that resulted in the final 522 Starships

- p34: add scale bar to Sup Fig1 panel b.

Done. X-axis scales were added to the boxplots. We did not include a scale bar for the cladogram, as it is intended to show tree topology rather than evolutionary distance, making a scale unnecessary. (see Supplementary Material, Supplementary Figure 1)

References within this response letter:

1. Gluck-Thaler, E. *et al.* Giant Starship Elements Mobilize Accessory Genes in Fungal Genomes. *Molecular Biology and Evolution* **39**, msac109 (2022).